# Synthesis and Evaluation of Some New 4*H*-Pyran Derivatives as Antioxidant, Antibacterial and Anti-HCT-116 Cells of CRC, with Molecular Docking, Antiproliferative, Apoptotic and ADME Investigations

**DOI:** 10.3390/ph15070891

**Published:** 2022-07-19

**Authors:** Nahed N. E. El-Sayed, Magdi E. A. Zaki, Sami A. Al-Hussain, Abir Ben Bacha, Malika Berredjem, Vijay H. Masand, Zainab M. Almarhoon, Hanaa S. Omar

**Affiliations:** 1National Organization for Drug Control and Research, Egyptian Drug Authority (EDA), 51 Wezaret El-Zeraa St., Giza 35521, Egypt; 2Department of Chemistry, Faculty of Sciences, Imam Mohammad Ibn Saud Islamic University, Riyadh 13318, Saudi Arabia; sahussain@imamu.edu.sa; 3Biochemistry Department, College of Sciences, King Saud University, Riyadh 11495, Saudi Arabia; aalghanouchi@ksu.edu.sa; 4Laboratory of Plant Biotechnology Applied to Crop Improvement, Faculty of Science of Sfax, University of Sfax, Sfax 3038, Tunisia; 5Laboratory of Applied Organic Chemistry LCOA, Synthesis of Biomolecules and Molecular Modeling Group, Badji-Mokhtar-Annaba University, Annaba 23000, Algeria; mberredjem@yahoo.fr; 6Department of Chemistry, Vidya Bharati College, Amravati 444602, Maharashtra, India; vijaymasand@gmail.com; 7Department of Chemistry, College of Science, King Saud University, Riyadh 11451, Saudi Arabia; zalmarhoon@ksu.edu.sa; 8Department of Genetics, Faculty of Agriculture, Cairo University, Giza 12613, Egypt; hanaa.omar@agr.cu.edu.eg; 9GMO Laboratory, Faculty of Agriculture, Cairo University, Research Park, Giza 12613, Egypt

**Keywords:** colorectal cancer, dysbiosis, oxidative stress, 4*H*-pyran, CDK2, caspases, ADME

## Abstract

Colorectal cancer oncogenesis is linked to dysbiosis, oxidative stress and overexpression of CDK2. The 4*H*-pyran scaffold is considered an antitumoral, antibacterial and antioxidant lead as well as a CDK2 inhibitor. Herein, certain 4*H*-pyran derivatives were evaluated as antibacterial, antioxidant and cytotoxic agents against HCT-116 cells. Derivatives **4g** and **4j** inhibited all the tested Gram-positive isolates, except for *B. cereus* (ATCC 14579), with lower IC_50_ values (µM) than ampicillin. In addition, **4g** and **4j** demonstrated the strongest DPPH scavenging and reducing potencies, with **4j** being more efficient than BHT. In cell viability assays, **4d** and **4k** suppressed the proliferation of HCT-116 cells, with the lowest IC_50_ values being 75.1 and 85.88 µM, respectively. The results of molecular docking simulations of **4d** and **4k**, inhibitory kinase assays against CDK2, along with determination of CDK2 protein concentration and the expression level of CDK2 gene in the lysates of HCT-116 treated cells, suggested that these analogues blocked the proliferation of HCT-116 cells by inhibiting kinase activity and downregulating expression levels of CDK2 protein and gene. Moreover, **4d** and **4k** were found to induce apoptosis in HCT-116 cells via activation of the caspase-3 gene. Lastly, compounds **4g**, **4j**, **4d** and **4k** were predicted to comply with Lipinski’s rule of five, and they are expected to possess excellent physiochemical and pharmacokinetic properties suitable for in vivo bioavailability, as predicted by the SwissADME web tool.

## 1. Introduction

Colorectal cancer (CRC) is a huge international health burden. It is currently ranked as the third most common cancer and the fourth most common cause of cancer-related death globally [1]. According to estimates from GLOBOCAN, in 2020 there were 1.9 million new CRC incident cases and 0.9 million deaths worldwide. The global projections for 2040 of new CRC cases are estimated to reach approximately 3.2 million [2].

Colorectal tumorigenesis is a highly complex process (Figure 1). Although little is known about the exact causes of sporadic cancer (with no family history of genetic predisposition), it is initiated by a number of carcinogenic events, which lead to the accumulation of genetic mutations in oncogenes and tumor suppressor genes, in addition to epigenetic modifications. These events drive the transformation of normal cells into uncontrolled adenomas and eventually to malignant carcinomas [3]. A good understanding of the factors promoting or accompanying these carcinogenic events would enable better policies for prevention and targeted therapy. Oxidative stress and dysbiosis of colon microflora constitute risk factors that drive these carcinogenic events, while deregulation of cell-cycle related proteins is a key feature of all cancer cells [4].

Oxidative stress is caused by an overwhelming production of reactive oxygen species (ROS) accompanied by the downregulation of antioxidant enzymes within the cell. These reactive molecular species may be generated from endogenous sources (intrinsic), including mitochondria, inflammatory cells and several enzymatic cellular complexes. In addition, they can originate from external sources (extrinsic), such as toxins, certain drugs, radiation and tobacco smoking. It is now well established that uncontrolled increased levels of reactive oxygen species contribute to the etiology of CRC via different mechanisms. One mechanism involves lipid peroxidation via attacking polyunsaturated fatty acids, which lead to the production of malonaldehyde, hexanal, acrolein and 4-hydroxy-2-nonenal, all of which are known mutagenic products capable of inducing persistent destabilization of chromosomes [5]. A second mechanism involves DAN modification, which results in mutations particularly in the p53 gene and mitochondrial DNA [5]. A third mechanism comprises induction of chronic inflammatory bowel diseases (IBDs), including, Crohn’s disease (CD) and ulcerative colitis (UC) [6]. Moreover, deregulation of ROS is accompanied by the generation of an inflammatory environment, which suppresses apoptosis and activates proliferation and angiogenesis, thus eventually leading to the initiation of neoplastic transformation [7]. In addition, several studies have also proved the involvement of ROS in the migration and invasion of cancerous cells [8]. Furthermore, ROS play a prominent role in chemotherapy resistance toward drugs, such as 5-fluorouracil, vinblastine, doxorubicin and tamoxifen [8]. Consequently, trapping free radicals with antioxidants seems to be a possible approach to prevent inflammation and cancer. Additionally, antioxidants are reported to exhibit anticancer activities through activating immune response, inhibiting angiogenesis, downregulating oncogenes and stimulating tumor suppressor genes [9].

Experimental evidence has established that dysbiosis of certain intestinal bacterial strains can promote chronic gastrointestinal inflammations, which eventually will lead to the development and spread of CRC [10,11]. The Gram-negative strains, *Escherichia coli* [12] and *Klebsiella pneumonia* belong to this class of virulent pathogens [13]. Additionally, certain studies have hypothesized that *Pseudomonas* bacteria are directly pro-oncogenic and promoters for CRC [14]. 

Additionally, animal model studies have shown that *Enterococcus faecalis* caused colon inflammation after infection, which promoted CRC occurrence [15]. In addition to this, some case studies indicated that the mean copy number of *Enterococcus faecalis* in people with colorectal cancer was significantly higher than in those with polyps and healthy people, which implies the ability of this strain to induce colorectal carcinogenesis [16].

Despite the recent advances in novel chemotherapy, such as immune checkpoint inhibitors, limitations due to chemotherapeutic resistance remain. Many preclinical and clinical studies have shown that microbiota modulate cancer treatment responses via influencing pharmacokinetic properties, such as metabolism, enzymatic degradation and pharmacodynamics (i.e., immunomodulation), in cancer therapy. Therefore, some new insights into cancer treatment are proposed based on manipulating the gut microbiota to augment cancer therapeutic responses [17]. 

Cancer patients who are undergoing anticancer treatment have severely weakened immune systems; thus, they are more vulnerable to bacterial infections and their life-threatening complications. A previous retrospective surveillance study revealed that the most common associated bacterial pathogens were *Klebsiella pneumonia* and *Enterococcus faecium* (bloodstream), *Pseudomonas aeruginosa* (respiratory tract), *Escherichia coli* (in urinary tract), *Staphylococcus*
*aureus* and *S. epidermidis* (skin and soft tissues) [18]. It is worth noting that some of these microbes are rather difficult to treat, as they are considered to be drug-resistant isolates [19]. Furthermore, several reports have associated antibiotic resistance with oxidative stress [20]. Therefore, molecules exerting antioxidant [21] and/or antibacterial [22] effects would be considered as attractive candidates that could provide protection against the development of CRC and its complications. 

Up to 85% of CRC cases exhibit chromosomal instability (CIN), featuring changes in chromosome numbers and structure. CIN affects the expression of tumor-associated genes and/or genes regulating cell-cycle division [23]. Decades of research in diverse fields, including cell biology, yeast genetics, biochemistry and genetic engineering, have identified the regulators of the cell division cycle [24]. Normally, the transitions of the cell cycle of the mammalian cell are strictly mediated by many cyclins and their effectors, cyclin dependent kinases (CDKs), which are a group of conserved serine/threonine protein kinases, each of which regulates a specific stage of the cell cycle [25]. In human cells, 20 CDKs and 29 cyclins have been identified. The CDK family is comprised of CDK1–7 and CDK14–18, which mediate cell-cycle transitions and cell division, whereas CDK8–13 and CDK19–20 regulate gene transcription [26]. Thus, any deregulation in the function or change in the level of these enzymes can result in the induction of tumorigenesis, including CRC [27]. In fact, previous reports have confirmed that the expression of CDK2 is dramatically increased in various colon cancer cell lines [28], including HCT-116, HCT-15 [29], LoVo and DLD-1 [30], compared with “normal” human colon epithelial cells (HCECs). Thus, targeting CDK2 has been recognized as a potential therapeutic approach for colon cancer treatment [31]. In addition, CDK2 inhibitors are likewise expected to have effectiveness in combination with other drugs or where synthetic lethality can be identified [32].

Taken together, the discovery, design and development of novel effective molecules for neutralizing free radicals and eradicating virulent bacterial strains and inhibiting CKD2 are to be considered as preventive and therapeutic approaches against CRC.

Consequently, in a quest to design new chemical compounds to meet all these health needs, the pyran scaffold can be considered a potential synthetic target. The 4*H*-Pyran motif is embedded in bioactive 4*H*-chromene derivatives which display antitumoral and antibacterial effects, such as compounds **Ia**,**b** [33], and antioxidant activities, such as derivatives **IIa**,**b** [34]. Moreover, compelling evidence has indicated that CDK2 is a valid anticancer target of pyran-containing compounds **III**–**VI**, as shown in Figure 2 [35,36,37], due to its pro-tumorigenic role via interference with cell division cycle.

Synthetically, 4*H*-pyrans and pyrano[2,3-*c*]pyrazoles derivatives are easily accessible, using a variety of catalysts [38,39] at ambient temperature, or under microwave or ultrasound irradiations [40,41,42]. These compounds are proven to display antibacterial [43,44,45], antitumoral [46,47,48] and antioxidant activities [49].

Based on the above-stated facts and in continuation of our research program of anticancer drug discovery, we designed and synthesized some new 4*H*-pyran and pyrano[2,3-*c*]pyrazole derivatives for in vitro biological screening as antioxidant, antibacterial and antiproliferative agents against HCT-116 cells of human colorectal cancer [50,51]. The underlying mechanism of the antiproliferative actions of the promising cytotoxic 4*H*-pyran derivatives was studied using molecular docking simulations in the ATP binding pocket of CDK2 and through the performance of a CDK2 inhibitory assay. Moreover, the concentration of CDK2 protein and the expression level of CDK2 gene in the lysates of HCT-116 cells treated with the cytotoxic candidates were measured and compared with positive and negative control experiments. Furthermore, quantitative real time PCR experiments for the caspase-3 gene were carried out to investigate the apoptotic activity of these candidates. Lastly, the expected physiochemical and pharmacokinetic properties of the active candidates were predicted using a Lipinski’s rule of five filter and BOILED-EGG and bioavailability radar charts.

## 2. Results and Discussion

### 2.1. Chemistry

Initially, the 4*H*-pyran templates **4a**–**m** were prepared using one-pot, tandem cascade Knoevenagel condensation/Michael addition and cyclization reactions of diverse aryl aldehydes **1a**–**k**, malononitrile **2** and various β-ketoesters **3a**–**c** in ethanolic piperidine solutions at room temperature. Similarly, the synthesis of pyrano[2,3-*c*]pyrazole derivatives **5a**,**b** was efficiently achieved via one-pot, four-component reactions of ethyl acetoacetate **3b**, hydrazine hydrate, aromatic aldehydes **1e**,**f** and malononitrile **2** under thermal heating and in the presence of ethanol as a solvent and triethyl amine as a catalyst (Figure 1).

The structures of all synthesized compounds were confirmed by their spectroscopic data, in addition to the results of elemental analyses. Thus, the IR spectra of derivatives **4a**–**m** showed the characteristic stretching absorption bands in the frequency range of ν_max_ (KBr)/cm^−1^ 3478~3416 (asymmetric vibration—NH_2_) and 3349~3197 (symmetric vibration—NH_2_), 2203~2188 (C≡N) and 1724~1689 (C=O); whereas the IR spectra of **5a**,**b** indicated the presence of the stretching absorption bands at ν_max_ (KBr)/cm^−1^ 3377 and 3309 (asymmetric vibration—NH_2_), 3381 and 3357 (symmetric vibration—NH_2_), 3159 and 3183 (NH) and 2170 and 2183 (C≡N).

The detailed ^1^HNMR and ^13^CNMR analytical data for derivatives **4a**–**m** and **5a**,**b** are presented in the experimental section. In addition, the mass spectrum (EI) of each compound displayed the anticipated molecular ion peak. The spectra of some representative examples are provided in the Appendix A.

Finally, the results of the elemental analyses for C, H and N of all the synthesized compounds were found to be within the permissible limits.

### 2.2. Biological Screening

#### 2.2.1. Antioxidant Assays

Antioxidants are substances capable of free radical scavenging, quenching singlet oxygen, inactivating peroxides and other reactive oxygen species (ROS), chelating metal ions, quenching secondary oxidation products and inhibiting pro-oxidative enzymes. Therefore, the antioxidant potential of a compound can be evaluated using a variety of chemical assays with different mechanisms [52]. Since the synthesized compounds are electron-rich, with substituents capable of electron donation and H-bond donation as well, in this study, two radical scavenging methods were employed, including the 1,1 diphenyl-2-picrylhydrazyl (DPPH) assay and the ferrous-reducing antioxidant capacity (FRAC) assay.

Initially, the DPPH radical scavenging potency assay was performed at various concentrations of the studied compounds ranging from 0.03 to 1 mg/mL. The IC_50_ values for DPPH were calculated from the plot of the scavenging effect against the compound concentration. 

The obtained results (Table 1) showed that the compounds **4g**, **4j**, **4l**, **4m** and **4d** exhibited the highest scavenging activity at 1 mg/mL, with scavenging potencies of 90.50, 88.00, 70.20, 69.00 and 65.25 %, respectively, as compared to 95.30 % for BHT. At the same concentration, the remaining compounds demonstrated poor to moderate DPPH radical scavenging activities ranging from 23.25 to 57.10%. Determination of the half-maximal inhibitory concentrations indicated that compounds **4g** and **4j** displayed the most promising free radical scavenging activities, with the lowest IC_50_ values of 0.329 and 0.1941 mM, respectively, as compared to BHT, which exhibited 0.245 mM.

Secondly, FRAC was determined over the concentration range of 0.0−1 mg/mL via monitoring Fe^3+^−Fe^2+^ transformation in the presence of the studied compounds and the reference antioxidant BHT to estimate the values of efficient concentration (EC_50_), expressed in mM. As depicted in Table 1, the pyrans **4g**, **4j** and BHT exhibited comparable tendencies with respect to their reducing power, with EC_50_ values of 0.072, 0.074 and 0.089 mM, respectively, whereas the rest of the tested compounds were less effective, with EC_50_ values in the range of 0.149−1.562 mM.

#### 2.2.2. Antibacterial Assays

The results for the in vitro antibacterial properties of the synthesized compounds against five Gram-positive and four Gram-negative human-pathogenic strains, which were assessed by both the agar diffusion method (based on inhibition zone (IZ)) and the broth dilution bioassay (for the determination of the half-maximal inhibitory concentration (IC_50_) values), using ampicillin as the reference antibiotic drug, are shown in Table 2 and Table 3.

Some of the tested compounds were found to be more effective against Gram-positive bacteria than Gram-negative isolates by producing larger IZs and lowering IC_50_ against certain strains. Reduced sensitivity against Gram-negative pathogens can be attributed to the restricted permeability of the chemical inhibitors across their cell walls due to the existence of an outer cell envelope, which acts as a selective barrier [53].

The results against Gram-positive strains identified compound **4g** as the most active agent against *B. subtilis* (ATCC 6633), with a lowered IC_50_ value (25.69 µM) and a bigger inhibition zone of 28.00 mm, compared to the results obtained for ampicillin: IC_50_ = 37.20 µM, IZ = 25.50 mm. Despite pyrans **4f**, **4i**, **4j**, **4l** and **4m** displaying lower IC_50_ values (ranging from 36.79 to 23.64 µM), their IZs were bigger, in the range of 21.50 (**4j**) to 9.50 mm (**4i**).

Considering *S. aureus* (ATCC 25923) and *S. epidermidis* (ATCC 14990), bigger IZs (mm) were displayed by **4g** (27.50, 29.75) and **4j** (25.25, 26.65) as compared to the results of 21.50, 22.50 obtained for ampicillin, respectively. In addition, these derivatives exhibited lowered IC_50_ (µM) values (**4g**: 27.78, 30.32) and (**4j**: 33.34, 33.34) as compared to 38.64 and 50.09 produced by ampicillin against these isolates, respectively. 

Moreover, derivative **4b** selectively exerted a larger IZ (23.50 mm) and a lower IC_50_ value (32.42 µM) against *S. epidermidis*, whereas compounds **4a**, **4c**, **4d**, **4e**, **4f**, **4l**, **4m** and **5b** inhibited it, with lowered IC_50_ values (µM) spanning from 50.05 (**5b**) to 26.69 (**4d**), though they displayed smaller inhibition zones than the reference, ranging from 6.60 (**5b**) to 20.50 (**4e**). Therefore, *S. epidermidis* can be considered the most sensible strain to the synthesized compounds. Similarly, compounds **4c**, **4d**, **4e**, **4f**, **4h**, **4l** and **4m** suppressed the growth of *S. aureus*, with lowered IC_50_ values spanning from 24.40 (**4d**) to 38.35 (**4f**) and smaller IZs ranging from 9.00 (**4d** and **4l**) to 17.50 (**4m** and **4h**).

With regard to *E. faecalis* (ATCC 29122), pyran **4g** and the standard drug produced equal inhibition zones of 24.50 mm; however, **4g** was capable of exhibiting a lowered IC_50_ value of 31.82 µM in comparison with that of 36.49 µM for ampicillin.

Contrarily, pyrans **4d**, **4f**, **4j** and **4m** showed lower inhibition zones, ranging from 14.50 (**4d**) to 23.20 mm (**4j**), and lower IC_50_ values, ranging from 28.16 (**4j**) to 35.28 µM (**4m**), than ampicillin.

Lastly, although the same compounds **4g** and **4j** exerted the biggest inhibition zones of 29.00 and 24.50 mm relative to that of 23.50 mm exerted by ampicillin against *B. cereus*, they exhibited higher IC_50_ values (29.42 and 27.63 µM, respectively) as compared to ampicillin (25.76 µM).

Collectively, **4g** showed promising antibacterial efficiency (lower IC_50_ values and bigger IZs) against all of the tested isolates as compared to the reference antibiotic, except for *B. cereus*, which was the microbe least sensitive to the synthesized compounds. Similarly, **4j** was very effective against *S. aureus* (ATCC 25923) and *S. epidermidis* (ATCC 14990).

Contrarily, the anti-Gram-negative evaluation revealed that none of the studied compounds was capable of inhibiting the microbes with bigger IZs or lower IC_50_ values, except for **4a**, which inhibited *P. aeruginosa* (ATCC 27853), with an IC_50_ of 37.6 µM, which is lower than those of ampicillin (41.50 µM) and **4j**, which suppressed *E. coli* (ATCC 25966) by a comparable concentration of 57.12 µM to that of ampicillin (57.96 µM). 

Furthermore, the lowest IC_50_ values against *K. pneumonia* (ATCC 700603*)* and *S. enteric* (ATCC 43972) were displayed by the derivatives **4g** (61.84 and 54.07 µM) and **4j** (48.35 and 49.68 µM) as compared to 32.91 and 41.50 µM, respectively, for ampicillin. 

In summary, although none of the tested compounds showed significant activities against Gram-negative microbes, the observed anti-Gram-positive potential of compounds **4g** and **4j** is of great significance. This is due to the fact that Gram-positive strains are reported to be common causes of bloodstream, skin, soft-tissue and intra-abdominal infections [54], which result in severe manifestations, such as sepsis [55] and/or endocarditis [56], especially in immunosuppressed cancer patients. Furthermore, recent reports have indicated that some of these bacterial isolates showed multidrug-resistant profiles [57], particularly *E. faecalis* and *S. aureus*, which amplify their serious complications and increase health care expenditures.

The observed broad-spectrum anti-Gram-positive profiles of **4g** and **4j** as compared to the other analogues can be attributed to their improved antioxidant capabilities. Indeed, the positive correlation between these two activities is well-documented. Catechin, which is a well-known potent free radical scavenger, has been reported to display anti-Gram positive and anti-Gram negative activities as well. Several mechanisms have been reported to account for the toxicity of catechin towards bacteria. Oxidative damage through membrane permeabilization was found to be the inhibitory mechanism of catechin against *B. subtilis* [58]. In addition, some experimental data showed a significant decrease in superoxide dismutase (SOD) and catalase (CAT) activity after treatment of the Gram-positive *S. aureus* strain with catechin at its minimum inhibitory concentration (MIC). These inhibitions could be attributed to the capability of catechin as an antioxidant agent to chelate the metal cofactors (Zn, Fe and Mn) at SOD catalytic sites [59]. Inhibiting superoxide dismutase would increase the bacterial sensitivity to the reactive oxygen species, resulting in inhibition of bacterial growth [60]. Other alternative proposals for the bactericidal actions of catechin include its ability to generate hydrogen peroxide, which causes cell membrane damage [61]. Hydrogen peroxide generation by antioxidants was reportedly attributable to antioxidant autoxidation in culture media [62]. In the view of the aforementioned facts, it can be suggested that the antibacterial effects of the antioxidant candidates **4g** and **4j** might be attributed to one or more of the indicated mechanisms, which could be investigated in future.

#### 2.2.3. Cell Viability Assays

The synthesized compounds were also examined at four concentrations (10, 25, 50 and 100 µg/mL) for their cytotoxic effects on human colorectal cancer HCT-116 cells after their incubation together for 24 h. The results, which were expressed in terms of percent of viable cells at each concentration as plotted in Figure 3, revealed that analogs **4g** and **4j** were not able to inhibit the growth of the cancerous cells, even at the highest used concentration (100 µg/mL). Conversely, they displayed the highest detectable percentages of viable cells: 86.0 and 94.0% at 100 µg/mL, respectively. Meanwhile, eleven compounds, namely, **4k**, **4b**, **4d**, **6b**, **4l**, **4m**, **4c**, **4e**, **5a**, **4h** and **4i**, exhibited potent cytotoxic activity at a concentration of 100 µg/mL by reducing cells viability to the extent of 11.0–32.5. The remaining compounds in this series, namely, **4a** and **4f**, displayed the least cytotoxic effects with percent cell viability measures of 46.0 and 55.5, respectively. 

Based on these results, the IC_50_ values for the promising cytotoxic candidates were determined, as shown in Table 4.

The calculated IC_50_ values (µM) ranged from 75.10 (**4d**) to 332.59. The smallest median concentrations were exhibited by compounds **4d** (75.10), **4k** (85.88), **4m** (89.33), **4l** (93.35) and **5b** (97.91).

#### 2.2.4. Investigation of the Underlying Mechanism of Action for the Antiproliferative Candidates

##### 2.2.4.1. Molecular Docking Simulations against CDK2 as a Potential Molecular Target

The molecular docking approach has become a powerful tool for identifying compounds showing potential anticancer activity against a specific target and with a selective inhibition mechanism [63]. Cancer is increasingly viewed as a cell-cycle disease. Cyclin-dependent kinases (CDKs) are a crucial protein family involved in cell proliferation through regulating the progression of the cell cycle and transcription. Animal model studies and molecular analyses of human tumors have indicated that many of these regulators are altered in cancer, particularly, CDK4, CDK6 and CDK2 and their substrates, which control progression through the G1/S phases of cell-cycle division. A literature survey confirmed that CKD2 is a valid target for 4*H*-pyran derivatives (Figure 2) with anticancer activities [33,34,35,36,37]. Therefore, to investigate whether or not the cellular mechanism by which the most active cytotoxic derivatives **4d** and **4k** suppressed the proliferation of HCT-116 cells is related to the inhibition of CDK2, the induced-fit docking approach in the ATP active site of the enzyme using the Glide program of Schrodinger-Maestro 11.2 was employed prior to the in vitro inhibitory kinase assays. The ATP-binding pocket of CDK2 is considered to be the site most commonly targeted by the kinase inhibitors. They compete with ATP to bind at the kinase site by forming hydrogen bonds with backbone amino acid residues and by establishing hydrophobic interactions with side chains of surrounding residues [64], leading to the inhibition of kinase phosphorylation and suppression of CDK2 hyperactivation, thus, holding back infinite cell proliferation [65].

In this study, the docking procedures were initially validated using redocking approach. Thus, the CDK2-inhibitor 4-[3-hydroxyanilino]-6,7-dimethoxyquinazoline (co-crystallized ligand, **DTQ**) was removed, after which it was redocked to reproduce the reported complex (PDB code: 1DI8). The significant residues in the active site of the protein are Lys33, Glu51 and Asp145. Figure 4 shows the superimposition of the predicted and the co-crystal structure of the reference inhibitor in the ATP active site of the protein. Moreover, the value of the root-mean-square deviation (RMSD) between the redocked conformation of **DTQ** and that observed in the X-ray crystallographic complex was found to be 0.34 Å, which is less than the cutoff value (2 Å) for the correct docking procedures. These observations confirmed that the redocked ligand was closely bound to the true conformation, indicating the reliability of the docking protocols. 

Afterwards, a comparative in silico molecular modeling study using the most active cytotoxic derivatives **4d** and **4k**, along with the least active analogue, **4f**, and the well-known CDK2 inhibitor **BMS-265246**, which was used in the in vitro inhibitory kinase assays, as well as the co-crystallized ligand (**DTQ**), was performed. Upon completion of each docking calculation, a maximum of 100 poses per ligand were generated and the final best-docked conformations were ranked using an XP Glide score function and Glide energy. 

According to the data presented in Table 5, the cocrystal ligand **DTQ** and **BMS-265246** showed better scoring functions than the synthesized compounds. Both **4d** and **4k** showed high affinity to the target protein, with **4d** demonstrating comparable binding strength (Glide energy = −48.700 Kcal/mol) to the co-crystal ligand (Glide energy = −49.122 Kcal/mol) and **BMS-265246** (Glide energy = −47.340 Kcal/mol), whereas the inactive analog **4f** exhibited the lowest affinity to the enzyme with the highest Glide energy value (−30.726 Kcal/mol). 

Lastly, the lowest-energy docked complexes of the co-crystallized ligand **DTQ**, the reference inhibitor **BMS-265246**, **4d**, **4****k** and **4f** with the protein were used to identify the crucial interactions for inhibition of CDK2. The ligand–protein interactions (hydrogen bonds, as well as hydrophobic interactions) are shown in Figure 5, Figure 6, Figure 7, Figure 8, Figure 9, Figure 10, Figure 11, Figure 12, Figure 13 and Figure 14.


**Docking pose for DTQ:**


Through examination of molecular docking, the pose of **DTQ** is shown to be very interesting, as it involves an active site water molecule as a bridge for H-bond establishment [66]. In addition to this, the phenolic OH and N-1 of the pyrimidine ring of **DTQ** are responsible for H-bond formation with Lys33 and Leu83, respectively (Figure 5 and Figure 6).

It is noteworthy that the H-bonding interactions with Lys33 and Leu83 on the protein backbone are important for potent inhibitory activity [67].


**Docking pose for BMS-**
**265246:**


The docking pose for **BMS-265246** indicates that it coordinated different residues of the enzyme due to hydrophobic and polar interactions.. The Leu83 residue and the NH of the pyrazole ring of **BMS-265246** are responsible for a strong H-bond formation with a distance of 2.15 Å. The N-2 atom of the same pyrazole ring interacted with Asp86 due to mild polar interactions. The 2,6-diflouro-4-methyl phenyl moiety established hydrophobic interactions with the nearby residues, viz. Ala144, Asp145 and Leu148 (Figure 7 and Figure 8). From these docking results, the obtained pose of **BMS-265246** established similar important molecular interactions, in particular, H-bonding with Leu83, as documented before [68].


**Docking pose for 4d**
**:**


This pyran derivative formed three hydrogen bonds—with amino acid residue ASP86 through its amino group as a donor, with amino acid GLN131 through the 3-methoxyl group as an acceptor and the last with ASN132 as an acceptor through the 5-methoxyl group. These binding interactions confirmed the importance of the methoxy groups at the *m*-positions (3,5-positions) as well as the amino group for further optimization. Additionally, this ligand showed lipophilic interactions with ILE10, VAL18, ILE10, PHE80 and LEU148. Moreover, the phenyl ring was involved in pi–pi stacking interactions with PHE80 (Figure 9 and Figure 10) 


**Docking pose for 4k**
**:**


Likewise, the analysis of the binding interactions between CDK2 and compound **4k** (Figure 11 and Figure 12) revealed the formation of two hydrogen bonds between the amino acid residues ASP86 and LYS129 in the active pocket and amino (as a donor) and nitrile (as an acceptor) substituents of **4k**. Additionally, the phenyl ring of the benzyl moiety showed lipophilic interactions with ILE10, VAL18, ILE10, PHE80, LEU148, LEU134 and VAL64. 


**Docking pose for 4f**
**:**


Contrarily, the inactive derivative **4f** did not form any hydrogen bonds inside the active site of CDK2. However, it showed lipophilic interactions with ILE10, VAL18, LEU83 and Val64, as shown in Figure 13 and Figure 14.

Overall, it can be concluded that there is a positive correlation between the docking simulations and the results of the cell viability assays, which support our claim that CDK2 is a potential target which could be responsible for the observed antiproliferative activities of compounds **4d** and **4k**. 

Moreover, the docking analyses of **4d**, **4k** and **4f** successfully provided information on possible structural modifications to improve kinase inhibitory efficiency. These analyses highlighted that an inhibitor for CDK2 requires a bulkier ring to be attached to the main scaffold (the pyran ring in the present case) through a flexible chain of two to three atoms. Furthermore, it should be present at the periphery of the molecule to penetrate well inside the ATP binding site to establish lipophilic and mild polar interactions [69]. In addition, docking analyses indicated that H-bond donors or acceptors on the central scaffold could substantially enhance binding with the receptor.

##### 2.2.4.2. In Vitro CDK2 Inhibitory Assays

To further substantiate the promising results of the docking analyses of pyrans **4d** and **4K** in the ATP active site of CDK2, their in vitro potencies to inhibit the enzyme activity were evaluated over a concentration range of 0.01−10 µM. The results of kinase assays (Figure 15) revealed that both pyrans strongly affected CDK2 activity in a dose-dependent manner. Thus, compound **4d**, at 0.01, 0.1, 1, or 10 µM, decreased CDK2 activity by 26.9, 41.5, 72.5 and 89%, respectively, while compound **4k** inhibited kinase efficiency by 21.9, 42.4, 66.3 and 83.6%, respectively, at the same concentrations. On the other hand, the positive control **BMS-265246** was found to be more effective and reduced CDK2 activity by 92.3% at the highest concentration used (10 µM). 

Moreover, the IC_50_ values for **4k**, **4d** and **BMS-265246** were deduced (Figure 16) and they were found to be 0.214, 0.143 and 0.036 µM, respectively.

Thus, our results for the kinase inhibitory assays are in agreement with the previously reported studies, which indicated that tropane–pyran hybrid structures could be considered as promising core scaffolds for developing new anticancer agents acting as CDK2 inhibitors [37].

It is worth noting that inhibition of CDK2 activity is considered to be a good approach for preventing chemotherapy-induced alopecia (CIA) by arresting the cell cycle without sensitization of the epithelium [70].

Collectively, the results of the docking simulations, the cell viability inhibition assays in HCT-116 colon cancer cells and the in vitro CDK2 inhibitory assays confirmed the potential of pyrans **4K** and **4d** as antiproliferative agents via the induction of cell-cycle arrest through CDK-2 inhibition.

##### 2.2.4.3. Further Mechanistic Studies via Quantitative Determination of the Concentration of CDK2 Protein and the Expression Profile of the CDK2 Gene in HCT-116 Cells Treated with the Cytotoxic Candidates

During the pathogenesis of cancer in humans, the enzymatic machinery, including human cyclins and their kinase (CDK) partners, which control the decisions to progress from a resting state (G0) to the cell cycle (G0-to-G1 transition) and/or to progress from the G1 (Gap-1) phase (in which the cell prepares for DNA synthesis) to the S phase (DNA synthesis phase), was found to be dysregulated, resulting in unrestrained proliferation. Furthermore, mutations have been observed in genes encoding CDKs; thus, the growth promoter and activator of G1 phase progression (the CDK2 gene) has been reported to be overexpressed in some human cancer subtypes, including colorectal, leukemia, pancreatic and renal carcinomas [24,71]. Moreover, overexpression of CDK2 in primary CRC tumors is linked to lymph node metastasis [72]. Therefore, different strategies for therapeutic intervention have been proposed based on inhibiting CDK activity and silencing the co-expressed genes in colon cancer cells to prevent them from proliferating and to induce their death [73]. 

Based on these facts, the effects of **4d** and **4k** on the expression levels of the CDK2 protein as well as the CDK2 gene were studied in a two-step protocol.

First, the commercial Human CDK2 ELISA Kit was used to estimate concentrations of the protein in the lysate of HCT-116 cells treated with **4d** and **4k**. The results were compared with the positive (cells treated with the reference CDK inhibitor, **BMS-265246**) and negative control experiments. The data presented in Figure 17 show that the concentration of CDK2 was estimated to be 7.466 ng/mL in the negative control experiment, while it was reduced to 2.229, 3.932 and 4.634 ng/mL in the positive control, **4k**- and **4d**-treated samples, respectively. These results implied that compounds **4d** and **4k** downregulated the expression of the CDK2 protein 1.9- and 1.6-fold, respectively, as compared to the negative control.

Second, the expression profiles of the CDK2 gene in HCT-116 cells treated with compounds **4d** and **4k** were determined. The results were compared with the positive control (cells treated with the well-known cytotoxic drug 5-fluorouracil) and the negative control (untreated cells). 

As shown in Figure 18, the CDK2 transcript levels were decreased significantly, 0.29-, 0.45- and 0.50-fold, by **5-FU**, **4K** and **4d**, respectively, as compared to the untreated cells after 24 h of the treatment. 

Accordingly, our results imply that pyrans **4****K** and **4d** might cause growth arrest in HCT-116 cancer cells via downregulation of CDK2 gene and protein as well. 

#### 2.2.5. Investigation of the Apoptotic Potential via Real-Time PCR Determination of the Expression Profile of the Caspase-3 Gene in HCT-116 Cells Treated with Pyrans **4d** and **4k**

Normally, apoptosis, or programmed cell death, is considered as a safeguard mechanism for managing stress and maintaining tissue homeostasis. It proceeds through a series of well-ordered biochemical cascades of events, which are regulated by a network of proteins. However, due to genetic and epigenetic mutations, deregulation of these proteins occurs, which results in evasion of apoptosis. This oncogenic transformation is a hallmark of human cancer, and it contributes to other tumorigenic events, including uncontrolled growth, accumulation of further genetic mutations, tumor angiogenesis and metastasis and chemotherapeutic resistance [74]. Therefore, reinitiating selective cancer cell death is a fundamental goal in anticancer drug development. Research in cancer biology has provided fundamental insights into the apoptotic pathways and their deregulation in colon cancer. Moreover, it has identified the molecular targets for reinitiating selective tumor cell death, including antiapoptotic Bcl-2 family proteins, apoptosis inhibitors and activators of the TRAIL death receptor signaling pathway, as well as other apoptotic markers, such as caspases. Among the caspases, which play critical roles in apoptosis, are the initiators, including caspases-8 or 9, and the effectors, including caspase-3 and caspase-7. After being activated by the initiators, the executioner caspases cleave numerous vital structural and regulatory proteins, leading to cell death [75]. Previous studies have demonstrated that expression level of caspase-3 is downregulated in cancer; therefore, enhancing its activity with natural and synthetic compounds was suggested as a possible strategy for cancer therapy [76,77]. In fact, pyran-containing, 4*H*-chromene derivatives, such as 3-amino-1-(4-fluorophenyl)-1*H*-benzo[*f*]chromene-2-carbonitrile [78] and 3-amino-1-(4-iodophenyl)-8-methoxy-1*H*-benzo[*f*]chromene-2-carbonitrile [79], were found to be effective against HT-29 cells of CRC and -MDA-MB-231 cells of breast cancer, respectively, via a 3-caspase-dependent apoptosis mechanism; therefore, we anticipated that our closely related pyran analogues **4d** and **4k** might be capase-3 inducers as well. Thus, the real-time quantitative PCR analyses of caspase-3-gene levels following treatment with compounds **4d** and **4K** at a concentration of 10 mg/mL for 24 h, were performed to investigate their potential as proapoptotic agents. 

As shown in Figure 19, the treated HCT-116 cultures showed significant expression of caspase-3 gene as compared to the positive (**5-FU**-treated) and the negative (pure HCT-116 cells) control cultures. The transcription levels of the caspase-3 gene in HCT-116 in treated cultures were increased 7.31-, 4.62- and 2.55-fold by **5-FU**, **4K** and **4d**, respectively, as compared to the negative control. These results indicated a concentration of 10 mg/mL for increasing the expression of caspase-3 gene and subsequent induction of mitochondrial apoptosis of the HCT-116 cells.

Overall, pyrans **4d** and **4k** can be considered interesting candidates to be subjected to further studies in order to investigate their usefulness in developing new anti-CRC agents capable of inhibiting the proliferation of HCT-116 cells through inducing cell-cycle arrest at the G1/S boundary by targeting CDK2 and enhancing the apoptosis of cancerous cells via activation of the caspase-3 gene.

### 2.3. Physicochemical and Pharmacokinetic Properties and Lipinski’s Rule of Five

Six physicochemical parameters, comprising Lipophilicity (LIPO), Size, Polarity (POLAR), Insolubility (INSOL), Unsaturation (UNSAT) and Flexibility (FLEX) were predicted for the tested compounds using the bioavailability radar chart [80], in which the pink-colored zone (Figure 20) indicates the suitability of physicochemical properties to have good in vivo bioavailability. 

The results indicated that, compounds **4g** and **4d** did not violate any parameters in the radar chart, while **4j** and **4k** derivatives violated only the UNSAT parameter, which is acceptable. The calculated log *p* values were intermediate (2.27 for **4g**, 2.99 for **4j**, 2.98 for **4d** and 2.80 for **4k**). Based on these data, it can be predicted that all the tested derivatives would have good bioavailability profiles, especially the molecules **4g** and **4d**.

The SwissADME server provides a BOILED-EGG chart to indicate human intestinal absorption (white part), blood–brain barrier penetration (yellow part) and the probability of the tested compound acting as a substrate (PGP^+^, blue color) or not a substate (PGP^−^, red color) for permeability glycoprotein (PGP). 

The results (summarized in Figure 21) showed that all the tested derivatives can be expected to be of good oral absorption, which is indicated by their presence in the white area. Moreover, they would not penetrate the blood–brain barrier (BBB, egg yolk area); thus, they would not cause CNS toxicity. Finally, the four compounds are red-colored, which means that they are not anticipated to be substrates for PGP; therefore, they would not suffer from restricted entry to the target cells through efflux pumps [81].

Eventually, the drug-likeness and good oral bioavailability characteristics for the studied compounds were predicted using the Lipinski’s rule of five filter [82]. As shown in Table 6 no compound did violate any parameter of the rule, which states that no violation or even one violation indicates expected good oral bioavailability. 

Although the in-silico predictions are promising, these results are preliminary and should be confirmed experimentally.

## 3. Materials and Methods

### 3.1. Chemistry

#### 3.1.1. General

All reagents were used as supplied commercially. The IR spectra were obtained using potassium bromide (KBr) discs and recorded on a PerkinElmer FTIR spectrophotometer, Spectrum BX 1000 in wave number (cm^−1^). Nuclear magnetic resonance spectroscopy was performed on a Bruker Avance 500 spectrometer, operating at 500 MH_Z_ for ^1^H and 125 MH_Z_ for ^13^C at 25 °C (Research Unit, College of Pharmacy, Prince Sattam Bin Abdulaziz University, Al-Kharj, KSA), or on an Eclipse 300 FT NMR spectrometer operating at 300 MHz spectrometer for ^1^H and at 75 MHz for ^13^C at 25 °C (College of Science, King Saud University, Riyadh, KSA). The chemical shifts are expressed in ppm and tetramethyl silane (TMS) was used as internal standard; mass spectra were recorded on a Shimadzu Qp-2010 Plus Mass Spectrometer using Ionization Mode: EI (Micro Analytical Center, Cairo University, Egypt). Percentages of C, H and N in the new compounds were evaluated at the regional center for Mycology and Biotechnology (RCMB), Al-Azhar University.

#### 3.1.2. General Procedures for the Synthesis of 6-Amino-5-cyano-4-(aryl)-2-methyl-4*H*-pyran-3-carboxylic Acid Esters **4a**–**m**

To an ethanolic mixture (20 mL) of benzaldehyde derivative **1a**–**k** (0.004 mol), namely, 2-formyl-benzoic acid, 3-methxy-2-nitro benzaldehyde, 3,4,5-trimethoxy benzaldehyde, biphenyl-3-carbaldehyde, 2-chloro-3-hydroxy-benzaldehyde, 2,4,6-trimethoxy benzaldehyde, 4-fluorobenzaldehyde, 4-methoxybenzaldehyde, 4-formyl-benzonitrile, 2-chloro-benzaldehyde, or 2,4-dichlorobenzaldehyde, and malononitrile **2** (0.004 mol, 0.264 g), piperidine (0.8 mL) and the appropriate β-ketoester derivative **3a**–**c** (0.004 mol) namely; methyl acetoacetate, ethyl acetoacetate or benzyl acetoacetate were added. The resulting reaction mixture in each was stirred at room temperature for 24 h, then it was worked up as described below to obtain the desired product.

##### 6-Amino-4-(2-carboxy-phenyl)-5-cyano-2-methyl-4*H*-pyran-3-carboxylic acid methyl ester **4a**

Filtration of the obtained reaction suspension afforded a yellow solid, which was washed with H_2_O/dil. HCl, air dried and recrystallized from EtOH to give shiny white crystals, yield (63%), m.p. 216–218 °C; ν_max_ (KBr)/cm^−1^ 3328 (broad OH and NH_2_), 3193 (CH-Ar), 2895 (CH-aliphatic), 2656, 2544, 2196 (CN), 1716 (C=O-ester), 1675 (C=O-carboxylic), 1605 (C=C), 1445 1405, 1377, 1325, 1264, 1226, 1182, 1124, 1068, 1042, 955, 876, 842, 803, 749, 728, 648, 624, 518, 442; *δ*_H_ (500 MHz; DMSO-*d_6_*) 7.73 (1H, dd, *J* = 7.7, 1.1 Hz, CH-Ar), 7.50 (1H, t, *J* = 7.3 Hz, CH-Ar), 7.29–7.25 (2H, m, 2 × CH-Ar), 6.85 (2H, br. s, NH_2_), 5.63 (1H, s, C^4^H-pyran), 3.46 (3H, s, O-CH_3_), 2.31 (3H, s, CH_3_); *δ*_C_ (125 MHz; DMSO-*d_6_*), 168.60 (C=O), 165.83 (C=O), 158.73, 157.04, 146.03, 131.95, 130.48, 129.36, 129.01, 126.39, 119.27, 107.59 (4 × CH-Ar, 2 × C_q_-Ar, 3 × C_q_-pyran, CN), 56.05 and 51.29 (C^5^-pyran and O-CH_3_), 32.72 (C^4^H-pyran), 18.40 (CH_3_); MS (EI) *m*/*z* (%) [M^+^] 314.85 (0.24) for C_16_H_14_N_2_O_5_, 268.90 (39.95), 252.85 (14.51), 236.90 (100.00), 208.90 (11.83), 192.90 (10.28), 133.00 (31.62), 116.05 (21.80), 105.00 (41.68), 89.00 (27.86), 77.05 (77.39), 66.00 (29.21), 57.05 (15.98). Anal. Calcd. for C_16_H_14_N_2_O_5_: C, 61.14; H, 4.49; N, 8.91. Found; C, 61.36; H, 4.70; N, 9.07.

##### 6-Amino-5-cyano-4-(3-methoxy-2-nitro-phenyl)-2-methyl-4*H*-pyran-3-carboxylic acid ethyl ester **4b**

The excess solvent was evaporated and the resulting yellow solid was washed with water, air dried and recrystallized from EtOH to afford a pure product as a pale yellow solid, yield (85%), m.p. 220–222 °C; ν_max_ (KBr)/cm^−1^ 3475 (asymmetric NH_2_), 3351 (symmetric NH_2_), 3168 and 3098 (CH-Ar), 2989, 2940, 2898 and 2847 (CH-aliphatic), 2192 (CN), 1679 (C=O), 1629 and 1586 (C=C), 1530 (NO_2_), 1478, 1449, 1372, 1332, 1277, 1210, 1122, 1056, 1018, 956, 908, 851, 802, 655, 613, 562, 512, 443; *δ*_H_ (500 MHz; DMSO-*d_6_*) 7.51 (1H, t, *J* = 8.2 Hz, CH-Ar), 7.18 (1H, d, *J* = 8.4 Hz, CH-Ar), 7.06 (2H, br. s, NH_2_), 6.88 (1H, d, *J* = 7.9 Hz, CH-Ar), 4.35 (1H, s, C^4^H-pyran), 3.96 (2H, q, *J* = 7.1 Hz, O-CH_2_-CH_3_), 3.87 (3H, s, O-CH_3_), 2.32 (3H, s, CH_3_), 0.99 (3H, t, *J* = 7.1 Hz, O-CH_2_-CH_3_); *δ*_C_ (125 MHz; DMSO-*d_6_*), 164.76 (C=O), 158.89, 158.08, 149.97, 139.63, 137.79, 131.78, 120.83, 118.79, 111.67, 105.54 (3 × CH-Ar, 3 × C_q_-Ar, 3 × C_q_-pyran, CN), 60.21 (O-CH_2_), 56.55 and 55.80 (C^5^-pyran and O-CH_3_), 34.43 (C^4^H-pyran), 18.22 (2 × CH_3_); MS (EI) *m*/*z* (%) [M^+^] 359.80 (0.40) for C_17_H_17_N_3_O_6_, 341.85 (35.07), 324.85 (5.56), 311.90 (6.19), 296.85 (4.09), 283.85 (8.98), 272.90 (12.99), 297.90 (43.20), 251.90 (11.35), 237.90 (10.52), 226.90 (34.93), 206.90 (11.00), 178.90 (12.93), 160.95 (11.26), 140.05 (6.60), 127.05 (9.20), 115.05 (20.68), 102.00 (23.05), 90.00 (29.84), 76.05 (69.38), 67.00 (100.00), 59.01 (38.59), 51.05 (33.51); Anal. Calcd. for C_17_H_17_N_3_O_6_: C, 56.82; H, 4.77; N, 11.69. Found: C, 57.11; H, 4.83; N, 11.97.

##### 6-Amino-5-cyano-4-(3-methoxy-2-nitro-phenyl)-2-methyl-4*H*-pyran-3-carboxylic acid benzyl ester **4c**

Addition of ice/H_2_O to the reaction mixture and filtration gave the crude product as a yellow precipitate which was air dried and recrystallized from EtOH/petroleum ether to obtain the pure product as a yellow powder, yield (64%), m.p. 190–192 °C; ν_max_ (KBr)/cm^−1^ 3434 (asymmetric NH_2_), 3321 (symmetric NH_2_), 3221; 3190 and 3090, 3031 (CH-Ar), 2942 and 2846 (CH-aliphatic), 2195 (CN), 1716 (C=O), 1673 and 1603 (C=C), 1528 (NO_2_), 1479, 1449, 1406, 1370, 1315, 1285, 1225, 1174, 1181, 1133, 1063, 963, 912, 847, 804, 756, 695, 662, 566, 524, 435; *δ*_H_ (300 MHz; DMSO-*d**_6_*) 7.55-7.46 (2H, m, 2 × CH-Ar), 7.30–7.17 (3H, m, 3 × CH-Ar), 7.08 (2H, br. s, NH_2_), 7.07-7.01 (2H, m, 2 × CH-Ar), 6.87 (1H, d, *J* = 7.5 Hz, CH-Ar), 5.07 and 4.98 (2H, AB q, *J* = 12.6 Hz, O-CH_2_-Ph), 4.39 (1H, s, C^4^H-pyran), 3.88 (3H, s, O-CH_3_), 2.32 (3H, s, CH_3_); *δ*_C_ (75 MHz; DMSO-*d**_6_*), 164.11 (C=O), 158.80, 158.06, 150.11, 139.42, 137.64, 135.98, 131.82, 128.22, 127.79, 127.48, 120.89, 118.67, 111.77, 105.49 (8 × CH-Ar, 4 × C_q_-Ar, 3 × C_q_-pyran, CN), 65.56 (O-CH_2_-Ph), 56.64, 55.89 (C^5^-pyran and O-CH_3_), 34.65 (C^4^H-pyran), 18.48 (CH_3_); MS (EI) *m**/z* (%) [M^+^ + 1] 422.10 (0.25) for C_22_H_19_N_3_O_6_, 404.00 (1.48), 360.10 (1.07), 343.05 (0.02), 309.10 (0.76), 291.10 (1.16), 283.05 (1.27), 268.05 (3.94), 252.05 (1.83), 227. 00 (4.78), 201.00 (2.05), 153.05 (0.43), 140.05 (0.78), 107.10 (2.60), 92.05 (8.02), 91.05 (100.00), 77.05 (3.46), 65.00 (12.07), 51.00 (3.09); Anal. Calcd. for C_22_H_19_N_3_O_6_: C, 62.70; H, 4.54; N, 9.97; O, 22.78. Found: C, 62.59; H, 4.68; N, 10.19.

##### 6-Amino-5-cyano-2-methyl-4-(3,4,5-trimethoxy-phenyl)-4*H*-pyran-3-carboxylic acid benzyl ester **4d**

Filtration of the resulting mixture afforded a yellowish-white powder. Recrystallization from MeOH/petroleum ether afforded shiny white crystals, yield (57%), m.p. 158–160 °C; ν_max_ (KBr)/cm^−1^ 3430 (asymmetric NH_2_), 3327 (symmetric NH_2_), 3181 (CH-Ar), 2940 and 2887 (CH-aliphatic), 2373, 2192 (CN), 1711 (C=O), 1689, 1641 and 1593 (C=C), 1503, 1463, 1427, 1378, 1322, 1252, 1222, 1120, 1059, 995, 888, 856, 802, 759, 719, 693, 662, 578, 515 cm^−1^; *δ*_H_ (300 MHz; DMSO-*d_6_*) 7.37–7.25 (3H, m, 3 × CH-Ar of benzyl group), 7.18–7.11 (2H, m, 2 × CH-Ar of benzyl group), 6.92 (2H, br. s, NH_2_), 6.37 (2H, s, 2 × CH-Ar of trimethoxyphenyl group), 5.10 and 5.02 (2H, ABq, *J* = 12.9 Hz, O-CH_2_-Ph), 4.32 (1H, s, C^4^H-pyran), 3.68 (6H, s, 2 × O-CH_3_), 3.64 (3H, s, O-CH_3_), 2.35 (3H, s, CH_3_); *δ*_C_ (75 MHz; DMSO-*d_6_*), 165.45 (C=O), 158.44, 157.36, 152.89, 140.52, 136.47, 135.89, 140.52, 136.47, 135.89, 128.27, 127.88, 127.56, 119.76, 106.73, 104.351 (7 × CH-Ar, 5 × C_q_-Ar, 3 × C_q_-pyran, CN), 65.77 (O-CH_2_-Ph), 59.99, 57.29 and 55.76 (C^5^-pyran and 3 × O-CH_3_), 39.30 (C^4^H-pyran, 18.30 (CH_3_); MS (EI) *m*/*z* (%) [M^+^ + 1] 437.05 (6.98) for C_24_H_24_N_2_O_6_, [M^+^] 436.05 (24.88), 345.00 (36.17), 327.00 (2.08), 301.05 (3.55), 284.00 (5.49), 269.05 (9.70), 244.00 (3.34), 221.00 (2.36), 168.00 (2.28), 134.05 (1.57), 128.05 (0.90), 109.05 (1.08), 91.05 (100.00), 65.00 (9.29), 51.00 (2.45). Anal. Calcd. for C_24_H_24_N_2_O_6_: C, 66.04; H, 5.54; N, 6.42. Found: C, 66.21; H, 5.59; N, 6.64.

##### 6-Amino-4-biphenyl-3-yl-5-cyano-2-methyl-4*H*-pyran-3-carboxylic acid ethyl ester **4e**

The resulting mixture was poured onto ice/H_2_O and filtered to give a pale-yellow powder, air dried and recrystallized from EtOH/ petroleum ether to afford white crystals, yield (73%); m.p. 186–190 °C; ν_max_ (KBr)/cm^−1^ 3374 (asymmetric NH_2_), 3213 (asymmetric NH_2_), 3060 and 3032 (CH-Ar), 2976 and 2922 (CH-aliphatic), 2368, 2340, 2188 (CN), 1708 (C=O), 1676 and 1602 (C=C), 1475, 1419, 1373, 1321, 1262, 1211, 1180, 1132, 1069, 951, 902, 868, 774, 746, 698, 647, 545, 513; *δ*_H_ (500 MHz; DMSO-*d**_6_*) 7.62 (2H, d, *J* = 7.5 Hz, 2 × CH-Ar), 7.52 (1H, d, *J* = 7.9 Hz, CH-Ar), 7.48 (2H, t, *J* = 7.55 Hz, 2 × CH-Ar), 7.44–7.41 (2H, m, 2 × CH-Ar), 7.38 (1H, t, *J* = 7.3 Hz, CH-Ar), 7.17 (1H, d, *J* = 7.8 Hz, CH-Ar), 6.98 (2H, br. s, NH_2_), 4.43 (1H, s, C^4^H-pyran), 3.98 (2H, q, *J* = 7.1 Hz, O-CH_2_-CH_3_), 2.35 (3H, s, CH_3_), 1.02 (3H, t, *J* = 7.1 Hz, O-CH_2_-CH_3_); *δ*_C_ (125 MHz; DMSO-*d**_6_*), 165.46 (C=O), 158.54, 156.81, 145.60, 140.21, 140.04, 129.19, 128.97, 127.48, 126.58, 126.21, 125.51, 125.28, 119.76, 107.07 (9 × CH-Ar, 3 × C_q_-Ar, 3 × C_q_-pyran, CN), 60.16 (O-CH_2_-Ph), 57.15 (C^5^-pyran), 38.85 (C^4^H-pyran), 18.22 (CH_3_), 13.67 (CH_3_); MS (EI) *m**/z* (%) [M^+^ + 1] 360.85 (9.65) and [M^+^] 359.90 (36.27) for C_22_H_20_N_2_O_3_, 330.85 (26.30), 313.90 (24.30), 293.90 (4.77), 286.90 (15.54), 270.90 (8.75), 228.90 (10.19), 206.90 (100.00), 178.90 (24.65), 160.95 (15.76), 133.05 (6.67), 115.00 (10.63), 105.05 (15.92), 88.00 (16.94), 77.05 (22.94), 67.00 (44.11), 51.00 (14.94); Anal. Calcd. for C_22_H_20_N_2_O_3_: C, 73.32; H, 5.59; N, 7.77. Found: C, 73.59; H, 5.71; N, 7.98.

##### 6-Amino-4-(2-chloro-3-hydroxy-phenyl)-5-cyano-2-methyl-4*H*-pyran-3-carboxylic acid methyl ester **4f**

The resulting mixture was poured on ice–dil. HCl mixture, then filtered and the obtained solid was air dried to obtain an off-white powder. Recrystallization from EtOH/petroleum ether gave a shiny white powder yield (85%), m.p. 238–240 °C; ν_max_ (KBr)/cm^−1^ 3478 (OH), 3319 (NH_2_), 3015 (CH-Ar), 2946 and 2843 (CH-aliphatic), 2366, 2200 (CN), 1713 (C=O), 1674, 1634 and 1593 (C=C), 1462, 1403,1374, 1349, 1261, 1227.7, 1187, 1133, 1074, 976, 853, 801, 759, 726, 667, 547, 500, 443; *δ*_H_ (500 MHz; DMSO-*d_6_*) 10.09 (1H, s, OH), 7.08 (1H, t, *J* = 7.9 Hz, CH-Ar), 6.85 (2H, br. s, NH_2_), 6.81 (1H, dd, *J* = 8.1, 1.1 Hz, CH-Ar), 6.60 (1H, dd, *J* = 7.6, 1.1 Hz, CH-Ar), 4.93 (1H, s, C^4^H-pyran), 3.49 (3H, s, O-CH_3_), 2.34 (3H, s, CH_3_); *δ*_C_ (125 MHz; DMSO-*d_6_*), 165.82 (C=O), 158.60, 157.52, 152.94, 143.39, 127.38, 119.63, 119.50, 119.30, 119.23, 114.55, 106.22 (3 × CH-Ar, 3 × C_q_-Ar, 3 × C_q_-pyran, CN), 56.45 and 51.41 (C^5^-pyran and O-CH_3_), 35.28 (C^4^H-pyran), 18.12 (CH_3_); MS (EI) *m*/*z* (%) [M^+^ + 1, ^37^Cl] 322.80 (0.92); [M^+^, ^37^Cl] 321.80 (4.70); [M^+^ + 1, ^35^Cl] 320.80 (3.09) and [M^+^, ^35^Cl] 319.80 (13.67) for C_15_H_13_ClN_2_O_4_, 306.80 (1.28), 304.80 (3.81), 285.85 (10.38), 284.90 (58.22), 272.85 (1.03), 270.85 (0.36), 252.85 (5.53), 246.85 (1.34), 244.85 (3.08), 224.90 (4.03), 222.90 (1.39), 193.90 (10.96), 192.90 (100.00), 175.90 (9.94), 160.95 (28.16), 127.05 (6.44), 114.00 (10.61), 105.05 (20.05), 99.00 (12.74), 89.00 (23.42), 77.05 (19.61), 67.00 (54.84), 59.05 (37.40), 57.05 (11.01), 51.05 (10.88). Anal. Calcd. for C_15_H_13_ClN_2_O_4_: C, 56.17; H, 4.09; N, 8.73. Found: C, 56.44; H, 4.32; N, 8.99.

##### 6-Amino-4-(2-chloro-3-hydroxy-phenyl)-5-cyano-2-methyl-4*H*-pyran-3-carboxylic acid ethyl ester **4g**

The resulting reaction mixture was poured onto ice–dil. HCl mixture, then filtered and the obtained solid was air dried to afford an off-white powder. Recrystallization from EtOH/petroleum ether gave a shiny off white powder (65%), m.p. 212–213 °C; ν_max_ (KBr)/cm^−1^ 3454 (OH), 3350 (NH_2_), 3197 (CH-Ar), 2974 and 2853 (CH-aliphatic), 2642, 2574, 2372, 2203 (CN), 1713 (C=O), 1669 and 1606 (C=C), 1583, 1465, 1408, 1375, 1322, 1293, 1234, 1185, 1134, 1073, 975, 862, 807, 763, 727, 670, 560, 487; *δ*_H_ (300 MHz; DMSO-*d_6_*) 10.10 (1H, s, OH), 7.08 (1H, t, *J* = 7.1 Hz, CH-Ar), 6.95 (2H, br. s, NH_2_), 6.83 (1H, d, *J* = 7.8 Hz, CH-Ar), 6.60 (1H, d, *J* = 7.8 Hz, CH-Ar), 4.90 (1H, s, C^4^H-pyran), 3.94 (2H, q, *J* = 7.2 Hz, O-CH_2_-CH_3_), 2.38 (3H, s, CH_3_), 0.98 (3H, t, *J* = 7.2 Hz, O-CH_2_-CH_3_); *δ*_C_ (75 MHz; DMSO-*d**_6_*), 165.28 (C=O), 158.55, 157.39, 152.98, 143.49, 127.42, 119.48, 119.29, 114.48, 106.37 (3 × CH-Ar, 3 × C_q_-Ar, 3 × C_q_-pyran, CN), 60.09 (O-CH_2_), 56.26 (C_q_-pyran), 34.6 (C^4^H-pyran), 18.08 (CH_3_), 13.62 (CH_3_); MS (EI) *m**/z* (%) [M^+^ + 1, ^37^Cl] 337.00 (0.63), [M^+^, ^37^Cl] 336.00 (3.06), [M^+^ + 1, ^35^Cl] 335.00 (2.34) and [M^+^, ^35^Cl] 334.00 (8.94) for C_1__6_H_1__5_ClN_2_O_4_, 307.00 (5.17), 305.00 (14.95), 300.05 (15.37), 299.10 (81.62), 291.00 92.77), 289.00 (7.87), 271.05 (3.09), 261.00 (9.05), 225.00 (6.39), 208.00 (12.99), 207.00 (100.00), 179.00 (31.94), 161.00 (17.35), 133.05 (6.00), 127.05 (4.38), 115.05 (3.79), 105.05 (5.32), 89.05 (6.04), 77.05 (3.88), 67.00 (15.90), 63.00 (11.37), 57.05 (3. 59), 51.00 (4.07); Anal. Calcd. for C_16_H_15_ClN_2_O_4_: C, 57.41; H, 4.52; N, 8.37. Found: C, 57.57; H, 4.41; N, 8.56.

##### 6-Amino-5-cyano-2-methyl-4-(2,4,6-trimethoxy-phenyl)-4*H*-pyran-3-carboxylic acid ethyl ester **4h**

Filtration of the resulting reaction mixture afforded a yellow solid, which was washed with water and air dried. Recrystallization from EtOH afforded the pure product as a yellow powder, yield (60%), m.p. 196–198 °C; ν_max_ (KBr)/cm^−1^ 3465 (asymmetric NH_2_), 3330 (symmetric NH_2_), 3228 and 3192 (CH-Ar), 2940 and 2839 (CH-aliphatic), 2371, 2198 (CN), 1709 (C=O), 1688 and 1602 (C=C), 1460, 1404, 1330, 1226, 1150, 1119, 1062, 948, 922, 816, 785, 717, 649, 606, 500; *δ*_H_ (300 MHz; DMSO-*d**_6_*) 6.54 (2H, br. s, NH_2_), 6.18 (2H, s, 2 × CH-Ar), 4.88 (1H, s, C^4^H-pyran), 3.90 (2H, q, *J* = 6.9 Hz, O-CH_2_-CH_3_), 3.75 (3H, s, O-CH_3_), 3.71 (6H, s, 2 × O-CH_3_), 2.23 (3H, s, CH_3_), 1.02 (3H, t, *J* = 6.9 Hz, O-CH_2_-CH_3_); *δ*_C_ (75 MHz; DMSO-*d**_6_*) 166.31 (C=O), 159.85, 159.700, 159.13, 156.94, 120.54, 112.39, 105.44, 91.41 (2 × CH-Ar, 4 × C_q_-Ar, 3 × C_q_-pyran, CN), 59.78, 56.36, 56.12, 55.96, 55.27 (O-CH_2_, C^5^-Pyran, 3 × O-CH_3_), 27.26 (C^4^H-pyran), 18.08 (CH_3_), 13.91 (CH_3_); MS (EI) *m**/z* (%) [M^+^ + 1] 375.00 (5.63) and [M^+^] 374.05 (24.69) for C_19_H_22_N_2_O_6_, 345.00 (17.76), 343.00 (29.18), 329.05 (6.42), 313.00 (4.86), 297.00 (100.00), 285.05 (18.39), 269.05 (13.20), 257.05 (21.80), 241.05 (4.48), 227.00 (4.35), 207.00 (11.32), 179.00 (12.32), 161.00 (11.25), 134.05 (10.03), 77.00 (8.03), 66.95 (15.11), 57.00 (5.43), 51.00 (3.59). Anal. Calcd. for C_19_H_22_N_2_O_6_: C, 60.95; H, 5.92; N, 7.48. Found: C, 61.17; H, 6.08; N, 7.71.

##### 6-Amino-5-cyano-4-(4-fluoro-phenyl)-2-methyl-4*H*-pyran-3-carboxylic acid benzyl ester **4i**

The resulting reaction mixture was poured onto iced water and the precipitated solid was filtered off to obtain a yellow powder, which was washed with plenty of water, air dried, recrystallized from EtOH/ petroleum ether to give a pale yellow powder, yield (80%), m.p. 124–126 °C; ν_max_ (KBr)/cm^−1^ 3458 (asymmetric NH_2_), 3324 (symmetric NH_2_), 3238 and 3095 (CH-Ar), 2939 (CH-aliphatic), 2370, 2199 (CN), 1721 (C=O), 1680 and 1602 (C=C), 1504, 1450, 1405, 1376, 1321, 1219, 1177, 1119, 1056, 958, 852, 763, 742, 696, 595, 521, 466; *δ*_H_ (300 MHz; DMSO-*d**_6_*) 7.31–7.27 (3H, m, 3 × CH-Ar), 7.13-7.09 (6H, m, 6 × CH-Ar), 6.94 (2H, br. s, NH_2_), 5.07 and 4.98 (2H, ABq, *J* = 16.4 Hz, O-CH_2_-Ph), 4.34 (1H, s, C^4^H-pyran), 2.32 (3H, s, CH_3_); *δ*_C_ (75 MHz; DMSO-*d**_6_*), 164.65 (C=O), 157.80, 156.91, 140.17, 135.43, 128.80, 128.72, 127.98, 127.63, 127.42, 115.06, 114.78, 106.18 (9 × CH-Ar, 3 × C_q_-Ar, 3 × C_q_-pyran, CN), 65.48 (O-CH_2_-Ph), 56.99 (C^5^-pyran), 38.74 (C^4^H-pyran), 18.04 (CH_3_); MS (EI) *m**/z* (%) [M^+^ + 1] 365.05 (0.42) and [M^+^] 364.05 (1.41) for C_21_H_17_FN_2_O_3_, 273.05 (44.27), 269.10 (9.28), 251.05 (0.71), 229.00 (3.41), 213.00 (2.77), 200.00 (0.74), 184.00 (1.45), 158.00 (1.24), 145.05 (1.10), 133.05 (1.33), 121.05 (0.83), 91.05 (100.00), 77.00 (2.43), 65.00 (13.54), 51.00 (3.00).

##### 6-Amino-5-cyano-4-(4-methoxy-phenyl)-2-methyl-4*H*-pyran-3-carboxylic acid benzyl ester **4j**

The resulting reaction mixture was poured onto iced water and the precipitate was filtered off to obtain a yellow solid, which was washed with plenty of water, air dried and recrystallized from EtOH–petroleum ether to give off-white shiny crystals, yield (87%), m.p. 170–172 °C; ν_max_ (KBr)/cm^−1^ 3407 (asymmetric NH_2_), 3328 (symmetric NH_2_), 3263, 3219 and 3009 (CH-Ar), 2958 (CH-aliphatic), 2370, 2189 (CN), 1698 (C=O), 1678, 1646 and 1603 (C=C), 1511, 1417, 1379, 1334, 1261, 1208, 1174, 1113, 1059, 1032, 1004, 948, 852, 813, 771, 692, 601, 568, 527; *δ*_H_ (300 MHz; DMSO-*d*_6_) 7.31–7.29 (3H, m, 3 × CH-Ar), 7.12–7.09 (2H, m, 2 × CH-Ar), 7.00 (2H, dd, *J* = 8.4, 2.7 Hz, 2 × CH-Ar), 6.94 (2H, br. s, NH_2_), 6.84 (2H, dd, *J* = 8.4, 2.72 Hz, 2 × CH-Ar), 5.05 and 5.3 (2H, ABq, *J* = 12.6 Hz, O-CH_2_-Ph), 4.27 (1H, s, C^4^H-pyran), 3.74 (3H, s, O-CH_3_), 2.31 (3H, s, CH_3_); *δ*_C_ (75 MHz; DMSO-*d**_6_*), 165.39 (C=O), 158.25, 158.14, 156.76, 136.82, 135.79, 128.24, 127.84, 127.63, 119.74, 113.83, 107.18 (9 × CH-Ar, 3 × C_q_-Ar, 4 × C_q_-pyran, CN), 65.67 (O-CH_2_-Ph), 57.57, 55.04 (C^5^-pyran and O-CH_3_), 38.92 (C^4^H-pyran), 18.23 (CH_3_); MS (EI) *m**/z* (%) [M^+^ + 1] 377.05 (0.99) and [M^+^] 376.05 (3.61) for C_22_H_20_N_2_O_4_, 310.05 (0.77), 285.05 (51.33), 269.05 (6.84), 241.05 (6.47), 225.00 (4.71), 212.00 (0.68), 184.00 (3.11), 161.00 (4.63), 135.05 (2.31), 108.05 (1.55), 91.05 (100.00), 77.00 (4.75), 65.00 (13.86), 51.00 (3.36). 

##### 6-Amino-5-cyano-4-(4-cyano-phenyl)-2-methyl-4*H*-pyran-3-carboxylic acid benzyl ester **4k**

Evaporation of the excess solvent afforded a sticky, dark-red residue, which was treated with iced water, to resolve it into a dark-red precipitate. Filtration, air drying and recrystallization from EtOH gave shiny golden crystals, yield (58%), m.p. 156–158 °C; ν_max_ (KBr)/cm^−1^ 3407 (asymmetric NH_2_), 3301 (symmetric NH_2_), 3265, 3224, 3196 and 3061 (CH-Ar), 2964 and 2887 (CH-aliphatic), 2373, 2203 (CN), 1720 (C=O), 1685 and 1607 (C=C), 1497, 1449, 1405, 1376, 1319, 1258, 1216, 1182, 1129, 1057, 913, 864, 797, 752, 696, 615, 549, 507, 484, 437; *δ*_H_ (300 MHz; DMSO-*d*_6_) 7.76 (2H, d, *J* = 8.1 Hz, 2 × CH-Ar), 7.33–7.27 (5H, m, 5 × CH-Ar), 7.08–7.02 (4H, m, 2 × CH-Ar and NH_2_), 5.08 and 4.96 (2H, ABq, *J* = 12.6 Hz, O-CH_2_-Ph), 4.44 (1H, s, C^4^H-pyran), 2.34 (3H, s, CH_3_); *δ*_C_ (75 MHz; DMSO-*d**_6_*), 164.98 (C=O), 158.50, 158.38, 150.35, 135.64, 132.57, 128.23, 127.92, 127.74, 119.31, 118.76, 109.61, 105.66 (9 × CH-Ar, 3 × C_q_-Ar, 3 × C_q_-pyran, 2 × CN), 65.79 (O-CH_2_-Ph), 56.33 (C^5^-pyran), 34.85 (C^4^H-pyran), 18.43 (CH_3_); MS (EI) *m**/z* (%) [M^+^] 371.00 (0.94) for C_22_H_17_N_3_O_3_, 280.00 (55.65), 269.05 (16.00), 236.00 (2.30), 178.95 (2.02), 128.05 (1.20), 107.05 (2.71), 91.05 (6.47), 65.00 (12.33), 51.00 (3.64). 

##### 6-Amino-4-(2-chloro-phenyl)-5-cyano-2-methyl-4*H*-pyran-3-carboxylic acid benzyl ester **4l**

Filtration of the resulting reaction suspension afforded a pale-yellow solid that was washed with plenty of water, air dried and recrystallized from MeOH–petroleum ether to give white crystals, yield (70%), m.p. 162–164 °C; ν_max_ (KBr)/cm^−1^ 3467 (asymmetric NH_2_), 3321 (symmetric NH_2_), 3217, 3180, 3058 and 3030 (CH-Ar), 2955 (CH-aliphatic), 2368, 2200 (CN), 1724 (C=O), 1678 and 1598 (C=C), 1467, 1440, 1402, 1378, 1321, 1229, 1177, 1127, 1060, 958, 830, 745, 698, 674, 616, 577, 499, 420; *δ*_H_ (300 MHz; DMSO-*d**_6_*) 7.37–7.14 (7H, m, 7 × CH-Ar), 7.02–6.90 (4H, m, 2 × CH-Ar and NH_2_), 5.07 and 4.98 (2H, ABq, *J* = 12.6 Hz, O-CH_2_-Ph), 4.91 (1H, s, C^4^H-pyran), 2.36 (3H, s, CH_3_); *δ*_C_ (75 MHz; DMSO-*d**_6_*), 165.34 (C=O), 158.59, 158.56, 140.06, 135.77, 132.06, 129.74, 129.49, 128.42, 128.27, 127.82, 127.57, 119.19, 105.76 (9 × CH-Ar, 3 × C_q_-Ar, 3 × C_q_-pyran, CN), 65.76 (O-CH_2_-Ph), 56.19 (C^5^-Pyran), 35.02 (C^4^H-pyran), 18.39 (CH_3_); MS (EI) *m**/z* (%) [M^+^, ^37^Cl] 382.05 (0.64) and [M^+^, ^35^Cl] 380.05 (1.17) for C_21_H_17_ClN_2_O_3_, 291.00 (8.32), 289.00 (24.11), 271.05 (0.47), 269.10 (12.27), 247.00 (0.67), 245.00 (3.25), 209.00 (1.63), 207.00 (0.46), 181.00 (1.10), 179.00 (0.92), 167.00 (0.95), 165.00 (1.23), 136.05 (0.29), 134.10 (0.90), 91.05 (100.00), 77.05 (3.48), 65.00 (16.45), 51.00 (4.41). Anal. Calcd. for C_21_H_17_ClN_2_O_3_: C, 66.23; H, 4.50; N, 7.36. Found: C, 66.48; H, 4.73; N, 7.57.

##### 6-Amino-5-cyano-4-(2,4-dichloro-phenyl)-2-methyl-4*H*-pyran-3-carboxylic acid benzyl ester **4m**

Filtration of the reaction suspension and recrystallization of the obtained solid from ethanol gave a white powder, yield (57%), m.p. 146–148 °C; ν_max_ (KBr)/cm^−1^ 3466 (asymmetric NH_2_), 3322 (symmetric NH_2_), 3223 and 3190 (CH-Ar), 2957 (CH-aliphatic), 2371, 2341, 2202 (CN), 1720 (C=O), 1683, 1642 and 1603 (C=C), 1462, 1402, 1375, 1223, 1176, 1124, 1098, 1060, 1005, 956, 844, 745, 698, 647, 584, 517, 455; *δ*_H_ (300 MHz; DMSO-*d_6_*) 7.45 (1H, d, *J* = 2.1 Hz, CH-Ar), 7.35 (1H, dd, *J* = 8.4, 2.1 Hz, CH-Ar), 7.28–7.18 (4H, m, 4 × CH-Ar), 7.05–6.93 (4H, m, 2 × CH-Ar, NH_2_), 5.09 (1H, d, *J* = 12.6 Hz, one of O-CH_2_-Ph), 4.92 (1H, d, *J* = 12.6 Hz, one of O-CH_2_-Ph), 4.87 (1H, s, C^4^H-pyran), 2.36 (3H, s, CH_3_); *δ*_C_ (75 MHz; DMSO-*d_6_*), 164.50 (C=O), 158.46, 158.09, 140.88, 135.27, 132.54, 131.57, 130.72, 128.37, 127.78, 127.50, 127.37, 118.61, 104.89 (8 × CH-Ar, 4 × C_q_-Ar, 3 × C_q_-pyran, CN), 65.37 (O-CH_2_), 55.30 (C^5^-pyran), 34.64 (C^4^H-pyran), 18.01 (CH_3_); MS (EI) *m*/*z* (%) [M^+^, ^37^Cl] 416.00 (0.58), [M^+^, ^35^Cl] 414.00 (0.79) for C_21_H_16_Cl_2_N_2_O_3_, 324.90 (19.35), 322.90 (29.45), 307.90 (0.29), 305.90 (0.37), 280.95 (1.65), 279.95 (0.60), 269.05 (13.68), 270.00 (4.80), 243.00 (1.44), 221.90 (0.76), 207.95 (1.20), 186.95 (0.94), 164.00 (1.00), 134.05 (1.62), 92.05 (8.02), 91.05 (100.00), 79.05 (2.57), 65.00 (14.26), 57.05 (5.88), 51.00 (3.42). Anal. Calcd. for C_21_H_16_Cl_2_N_2_O_3_: C, 60.74; H, 3.88; N, 6.75. Found: C, 60.95; H, 4.12; N, 6.93.

#### 3.1.3. General Procedures for Synthesis of 6-Amino-4-(aryl)-3-methyl-2,4-dihydro-pyrano[2,3-*c*]pyrazole-5-carbonitrile **5a**,**b**

To a mixture of ethyl acetoacetate **3b** (0.003, 0.39 g, 0.38 mL) and hydrazine hydrate (2.1 equiv., 0.0062, 0.31 g, 0.3 mL), the appropriate aromatic aldehyde **1e** or **1f** (0.003 mol) was added, followed by malononitrile **2** (0.003, 0.2 g), EtOH (20 mL) and Et_3_N (0.0007 mol, 0.07 g, 0.1 mL). The resulting reaction mixture was heated under reflux for 12 h. 

##### 6-Amino-4-(2-chloro-3-hydroxy-phenyl)-3-methyl-2,4-dihydro-pyrano[2,3-*c*]pyrazole-5-carbonitrile **5a**

Evaporation of the excess solvent and recrystallization of the remaining product from EtOH afforded a beige solid, yield (65%), m.p. 240–242 °C; ν_max_ (KBr)/cm^−1^ 3377 and 3309 (OH, NH_2_ and NH), 3159 (CH-Ar), 2929 (CH-aliphatic), 2698, 2368, 2171 (CN), 1650 and 1597 (C=C), 1515, 1486, 1458, 1412, 1344, 1291, 1257, 1189, 1161, 1102, 1051, 962, 863, 834, 782, 724, 670, 604, 559, 462; *δ*_H_ (300 MHz; DMSO-*d**_6_*) 12.12 (1H, s, NH), 10.18 (1H, br. s, OH), 7.11 (1H, t, *J* = 7.8 Hz, Ar-H), 6.90 (2H, br. s, NH_2_), 6.84 (1H, dd, *J* = 8.0, 1.1 Hz, Ar-H), 6.58 (1H, apparent d, *J* = 7.5 Hz, Ar-H), 5.06 (1H, s, C^4^H-pyran), 1.82 (3H, s, CH_3_); *δ*_C_ (75 MHz; DMSO-*d**_6_*) 161.19, 154.92, 153.01, 135.28, 127.4, 120.47, 119.26, 114.52, 97.57, (3 × CH-Ar, 3 × C_q_-Ar, 3 × C_q_-pyran, C_q_-pyrazole, CN), 55.99 (C^5^-pyran), 30.67 (C^4^H-pyran), 9.55 (CH_3_); MS (EI) *m**/z* (%) [M^+^ + 1 ^37^Cl] 304.95 (1.37), [M^+^, ^37^Cl], 304.00 (7.36), [M^+^ +1 ^35^Cl] 303.00 (5.84) and [M^+^, ^35^Cl] 302.00 (22.49) for C_14_H_11_ClN_4_O_2_, 278.00 (1.13), 276.00 (3.31), 269.05 (0.21), 267.05 (1.22), 237.00 (1.99), 201.00 (18.55), 176.00 (10.65), 175.00 (100.00), 128.05 (1.25), 115.10 (5.92), 105.05 (1.99), 99.00 (2.82), 89.00 (4.58), 77.00 (3.02), 66.00 (7.52), 63.00 (8.23), 57.05 (3.77), 51.00 (4.44). Anal. Calcd. for C_14_H_11_ClN_4_O_2_: C, 55.55; H, 3.66; N, 18.51. Found: C, 55.79; H, 3.78; N, 18.68.

##### 6-Amino-3-methyl-4-(2,4,6-trimethoxy-phenyl)-2,4-dihydro-pyrano[2,3-*c*]pyrazole-5-carbonitrile **5b**

The resulting reaction mixture was cooled down and poured onto ice/water to obtain a pale-yellow precipitate. Recrystallization from EtOH/DMF afforded a yellow/mustard solid (70%), m.p. 182–184 °C; ν_max_ (KBr)/cm^−1^ 3183 and 3183 (NH_2_ and NH), 3002 (CH-Ar), 2940 and 2841 (CH-aliphatic), 2696, 2372, 2184 (CN), 1655 and 1599 (C=C), 1490, 1462, 1409, 1334, 1281, 1226, 1149, 1118, 1040, 951, 922, 828, 759, 733, 650, 581, 526; *δ*_H_ (300 MHz; DMSO-*d**_6_*) 11.75 (1H, s, NH), 6.54 (2H, s, 2 × Ar-H), 6.14 (2H, br. s, NH_2_), 5.02 (1H, s, CH-pyran), 3.72 (3H, s, OCH_3_), 3.64 (6H, s, 2 × OCH_3_), 1.86 (3H, s, CH_3_); *δ*_C_ (75 MHz; DMSO-*d**_6_*) 161.68, 159.29, 155.09, 133.60, 120.28, 111.57, 97.63, (2 × CH-Ar, 4 × C_q_-Ar, 3 × C_q_-pyran, C^3^-pyrazole, CN), 55.80, 55.24, 54.79 (C^5^-pyran and 3 × O-CH_3_), 24.3 (C^4^H-pyran), 8.89 (CH_3_); MS (EI) *m**/z* (%) [M^+^] 302.00 (22.49) for C_17_H_18_N_4_O_4_, 342.05 (15.25), 327.05 (29.93), 311.00 (100.00), 295.00 (5.77), 276.05 (14.53), 245.00 (54.35), 230.00 (8.58), 201.00 (4.44), 175.00 (26.46), 139.10 (8.45), 121.05 (6.39), 98.05 (11.78), 77.05 (10.21), 65.05 (7.86), 57.05 (15.75). 

### 3.2. Biology

#### 3.2.1. Antioxidant Assays

The in vitro antioxidant potential of the synthesized compounds was assessed by two different techniques and compared to butylated hydroxytoluene (BHT), a widely used commercial antioxidant which was taken as the positive control.

##### DPPH Radical Scavenging Assay

The compounds’ capacities to scavenge the stable radical 1,1 diphenyl-2-picrylhydrazyl (DPPH) formed in solution by donation of a hydrogen atom or an electron was investigated according to the method described by Bersuder and coworkers [83]. Due to the fact that the initial blue/purple solution of diphenyl picrylhydrazine changes to yellow in the presence of compounds with a capacity to scavenge DPPH free radicals, this reaction is used as a measure of a compound’s ability to scavenge any free radical. Briefly, a 0.5 mL volume of DPPH ethanolic solution was mixed with an equal volume of each sample concentration (0.03 to 1 mg/mL), shaken strongly and incubated at room temperature for 1 h in darkness. The absorbance of the residual DPPH radicals was measured at 519 nm and compared to that of the control (without the tested compound). DPPH radical scavenging was calculated using the following formula: Scavenging effect (%) = (1 − A_compound_/A_Control_) × 100, where A_compound_ and A_control_ are the absorbances of the tested compound and of the control, respectively. A plot of the scavenging effect (%) versus the sample concentration was also performed to determine the compound concentration providing 50% inhibition (IC_50_). 

##### Reducing Power Assay

The reducing power of the studied compounds was assessed according to Oyaizu’s method [84]. Briefly, different concentrations (ranging from 0 to 1 mg/mL) of each tested compound were first mixed with 1 mL of 0.2 M sodium phosphate buffer (pH 6.6) and 1 mL of 1% potassium ferricyanide (K_3_Fe(CN)_6_) and incubated at 50 °C for 20 min. After the addition of 1.25 mL of 20% trichloroacetic acid (TCA), the mixture was centrifuged for 10 min at 3000 rpm and the upper layer solution was then mixed with 0.5 mL of 0.1% fresh ferric chloride and an equal volume of deionized water. Finally, the absorption of the resulting mixture was measured at 700 nm using a UV spectrophotometer against distilled water as the blank, while butylated hydroxytoluene (BHT) was used as a positive control. The sample ferric reducing power capability was indicated by increased absorbance. 

#### 3.2.2. Antibacterial Activity

The antibacterial activity of all synthesized compounds was evaluated against the Gram-negative strains *E. coli* (ATCC 25966), *K. pneumonia* (ATCC 700603), *P. aeruginosa* (ATCC 27853) and *S. enteric* (ATCC 43972) and against the Gram-positive strains *B. cereus* (ATCC 14579), *B. subtilis* (ATCC 6633), *E. faecalis* (ATCC 29122), *S. aureus* (ATCC 25923) and *S. epidermidis* (ATCC 14990), using the agar diffusion method through measuring the appearance of the inhibition zone on the surface of the top agar, as reported by Berghe and Vlietinck [85]. Ampicillin (10 µg/well) was used as the positive reference standard. Bacterial viability was also investigated by determining the colony-forming ability (CFU) of bacteria incubated at different time intervals without or with appropriate amounts of the compound, which were mixed with 2 ×10^7^ CFU/mL in sterile BHI and incubated under shaking for 60 min at 37 °C. Samples were serially diluted into sterile BHI, streaked onto media agar plates and incubated for 24 h at 37 °C. The antibacterial potency of the tested compounds was expressed as the residual number of CFUs with reference to the initial inoculums. The results presented as the half-maximal (50 %) inhibitory concentration (IC_50_) values are the means of three different measurements. 

#### 3.2.3. Cell Culture

Cytotoxic potency was examined for a human colon cancer cell line HCT-116 (American Type Culture Collection; Manassas, VA, USA) using various amounts of each compound (10, 25, 50 and 100 µg). Samples were diluted in Dulbecco’s Modified Eagle’s Medium, consisting of 10% Fetal Bovine Serum, and added to cells grown and cultured for 24 h in a 5% CO_2_-humidified incubator at 37 °C. Then, the activity of lactate dehydrogenase released from damaged cells was determined in the collected supernatant aliquots using an ELISA endpoint assay (Benchmark Plus, Bio-Rad, Hercules, CA, USA). As positive and negative controls, respectively, 0.1% Triton X-100 in the assay medium and the assay medium only were used. Cell viability, shown as mean values ± SDs (n = 3), was expressed as a relative percentage of the OD values determined at 600 nm in compound-treated cells and the control.

#### 3.2.4. CDK2 Inhibitory Assay

CDK-2 inhibition activity of the studied compounds **4d** and **4k** was evaluated using a commercial CDK2 ELISA Kit (cat. no.: 79599; BPS Biosciences, San Diego, CA, USA) following the manufacturer’s instructions. Briefly, different concentrations of each compound or the positive control **BMS-265246** (0.01, 0. 1, 1 or 10 µM) were incubated with 20 µL of diluted CDK2/CyclinA2 enzyme at 30 °C for 45 min. Following the addition of 50 µL Kinase-Glo^®^ Reagent to each well, the plate was incubated at room temperature for 15 min. Then, the luminescence signal was measured using the microplate reader. The CDK2 inhibitory activity was expressed as inhibition percentage, which was determined by comparison with a control experiment for comparative purposes. IC_50_ values were deduced from the curves. All measurements were performed in triplicate.

#### 3.2.5. In Vitro Quantitative Determination of CDK2 Concentration in HCT-116 Cells

The in vitro quantitative measurement of concentration of CDK2 in lysates of HCT-116 cells was carried out using a commercial Human CDK2 ELISA Kit (cat. no.: LS-F22176; LifeSpan Biosciences Inc., Seattle, WA, USA) in the presence or absence of the studied compounds (**4d** and **4k**), following the manufacturer’s instructions. Briefly, standards, blanks or samples were first incubated in the corresponding wells for 90 min at 37 °C. Then, biotinylated detection antibody was added to each well and the plate was incubated for 1 h at 37 °C followed by the addition of an Avidin–Horseradish Peroxidase (HRP) conjugate which binds to the biotin. After incubation of the mixture for 30 min at 37 °C, unbound Avidin–HRP conjugate was washed away and a TMB substrate was then added which reacts with the HRP enzyme, resulting in color development. After that, the reaction was stopped using a sulfuric acid stop solution to terminate the color-development reaction and the optical density of each well was measured using a microplate reader at 450 nm. **BMS-265246** was used as the standard drug for inhibition of CDK2. All measurements were performed in triplicate.

#### 3.2.6. Gene Expression Profiles

##### Design of the Primer

Primers specific for the cyclin-dependent kinase-2 (CDK2) and caspase-3 genes were designed using primer blast (https://www.ncbi.nlm.nih.gov/tools/primer-blast/, accessed on 31 October 2021) or primer3 (https://primer3.ut.ee/, accessed on 31 October 2021) software. The selected genes covered the two main groups of the cyclin-dependent kinase-2 (CDK2) gene and the caspase-3 gene. Primers were used in RT-qPCR analysis to amplify fragments of 100–200 bp in length (Table 7).

##### RNA Isolation and Reverse Transcription

Total RNA was extracted from each cell culture flask using the RNeasy extraction kit (RNeasy micro kit, cat. no. 74004). Up to 1 × 10^6^ cells, depending on the cell line, were disrupted in buffer RLT and homogenized and disrupted. Ethanol was then added to the lysate, creating conditions that promote the selective binding of RNA to the RNeasy membrane. The samples were then applied to the RNeasy Mini spin column. Total RNA binded to the membrane, contaminants were efficiently washed away and high-quality RNA was eluted in RNase-free water. All bind, wash and elution steps were performed by centrifugation in a micro-centrifuge, with DNAse I treatment. The amount of extracted RNA was quantified by estimating the absorbance at 260 nm. The purity of the RNA was checked by measuring the ratio of the absorbance at 260 and 280 nm, where a ratio ranging from 1.8 to 2.0 was taken to be pure. The absence of degradation of the RNA was verified by RNA electrophoresis on a 1.5% agarose gel containing ethidium bromide. First-strand cDNA was generated from 1 μg of each flask using the High-Capacity cDNA Archive Kit, Model for One-Step RT-PCR procedures (RT-PCR kit- BioRad-USA, cat. no. 345-0412), according to the manufacturer’s protocol.

##### Quantitative Real-Time PCR (qRT-PCR)

Quantitative real-time PCR was carried out with the Thermal Cycler Rotorgene Real-Time System II (Rotorgene, South Korea) with the SYBR kit. The primer sequences are shown in Table 5. The PCR reaction was carried out in triplicate in 96-well plates. The mixture included 12.5 μL SYBR premix ExTag, 1 μL of 60 ng cDNA as the template, 5 μL of 2 μmol/L primer premix and 6.5 μL of DNase-free nuclease water at a total volume of 25 μL. The thermal profile of the real-time system was one step at 95 °C for 30 s, followed by 30 to 45 cycles at 95 °C for 10 s (denaturation) and at 55 °C for 30 s (annealing and extension), followed by an added dissociation pattern. The actin gene was used as an internal control gene, was abundant and remained constant, and GAPDH was used as an internal standard (housekeeping gene).

##### Data Analysis

The relative expression levels through the average cycle threshold (CT) were successfully detected. Average CT values were calculated from the triplicate experiment conducted for each gene; the CT value was detected by subtracting the average CT value of genes from the CT value of actin and GAPDH genes. The relative expression levels of the target genes were calculated using the ΔΔ Ct method [86] and the reference genes GAPDH for the cancer cell line. Finally, a fold change equation (2^−1^) was used to estimate relative expression levels, while the standard deviation was calculated from the replicated experimental data.

### 3.3. In Silico Studies

The cyclin-dependent kinase 2 (CDK2) enzyme was selected as the molecular target to investigate the underlying molecular mechanism of the antiproliferative effects based on similarity to previously reported antitumoral pyran derivatives, which induced cell-cycle arrest via targeting CDK2 enzyme [35,36,37]. The structure of the target protein was retrieved from the Protein Data Bank (www.rcsb.org accessed on 10 January 2022). Thus, the three-dimensional crystal structure (PDB code: 1DI8) of CDK2 in complex with 4-[3-hydroxy anilino]-6,7-dimethoxyquinazoline (**DTQ**) was used. 

The protein was prepared with a protein preparation wizard available in the Schrodinger suite, using the standard protocol recommended by Schrodinger, which involved removal of water/solvent molecules, fixing of non-standard residues and adding hydrogens and partial charges. 

The structures of the studied compounds (ligands), including **4d**, **4f**, **4k**, **BMS-265246** and **DTQ**, were sketched using the built-in panel of Maestro 11.

Ligand preparation is a capability of the Schrodinger software suite that combines tools for generating 3D structures and 2D (SDF) representations as well as performing ligand geometry minimization. By employing the Ligprep protocol, all the ligands were organized using OPLS3 with default settings, and the output file was saved in maegz format automatically [87]. 

The Glide module [88] in extra precision (XP) mode [89] was used for flexible molecular docking of all the molecules inside the active site (ATP binding site of the kinase) [90], using default settings, as recommended in the manual for the Glide module from Schrodinger. The residues Lys33, Glu51 and Asp145, constituting the active site, were set as flexible residues. For validation of the docking protocol, the co-crystallized ligand **DTQ** (4-[3-hydroxyanilino]-6,7-dimethoxyquinazoline) was redocked, with an RMSD of 0.34 Å. 

Upon completion of each docking calculation, a maximum of 100 poses per ligand were generated, which were passed through a series of filters, and the final best docked structures were ranked using a Glide score function and Glide energy. The Glide score of the predicted poses, which is the scoring energy for the best pose (lowest energy) of each ligand in the binding site, and the Glide energy were used to quantify the binding strength of the different compounds to the target protein. The protein–ligand complexes were analyzed to examine various types of interactions. For the best-scored ligands, the 2D and 3D plots of molecular ligand–receptor interactions were analyzed for hydrogen bonds and halogen bonds.

#### ADME Evaluation

SwissADME [91] is a free web tool for evaluating the pharmacokinetics, drug-likeness and medicinal chemistry friendliness of small molecules. 

It was accessed on 10 January 2022 to predict the ADME properties of the bioactive synthesized compounds **4g**, **4j**, **4d** and **4k**.

## 4. Conclusions

Eleven new derivatives of 2-amino-4*H*-pyran and one 2,4-dihydropyrano[2,3-*c*]pyrazole were synthesized, fully characterized and examined for antioxidant and antibacterial activities as well as cytotoxicity on a HCT-116 cell line of colorectal cancer. Derivatives **4g** and **4j** showed promising antioxidant potencies as compared to BHT. Moreover**,** these analogues were more potent than ampicillin, displaying lowered IC_50_ values against the Gram-positive strains *B. subtilis* (ATCC 6633), *S. aureus* (ATCC 25923) and *S. epidermidis* (ATCC 14990) and *E. faecalis* (ATCC 29122). The cell viability assays showed that compounds **4d** and **4k** exhibited the strongest antiproliferative activities, whereas **4g** and **4j** were found to be inactive. Thus, the latter analogues would be suitable candidates for further toxicity studies to evaluate their potential as safe antioxidant and anti-Gram-positive bacterial agents that could be used as preventive and adjuvant therapeutic agents against CRC. The antiproliferative mechanism of action of **4d** and **4k** was investigated using molecular docking simulations within the ATP binding site of CDK-2. The docking results revealed that **4d** and **4k** would inhibit CDK2 activity by competing with ATP to bind on the kinase site through establishing hydrogen bonds with backbone amino acid residues and by establishing hydrophobic interactions with side chains of surrounding residues. In addition, the docking results revealed that a more powerful inhibitor for CDK2 requires a bulkier ring to be attached to the pyran ring through a flexible chain of two to three atoms. Moreover, H-bond donors or acceptors on the central scaffold could substantially enhance binding with the receptor. Further mechanistic studies, including kinase inhibitory assays, quantitative measurement of CDK-2 protein and real-time PCR profiling of CDK-2 gene in HCT-116 treated cells, confirmed that the antiproliferative actions of **4d** and **4k** could be attributed to inhibiting the activity and downregulating the expression level of CDK-2 protein and gene as well. Consequently, these derivatives would be useful leads for the generation of new anti-CRC agents that do not cause chemotherapy-induced alopecia and arrest the cell cycle without sensitization of the epithelium. Finally, investigation of the proapoptotic potential of these analogues using real-time PCR profiling of the caspase-3 gene in HCT-116 treated cells indicated that the concentration of 10 mg/mL of **4d** or **4k** is optimal for inducing mitochondrial apoptosis of the HCT-116 cells via upregulating the expression of caspase-3; therefore, further studies on this topic will be pursued in the future. Despite the in-silico predictions of the ADME profiles and drug-likeness properties of the bioactive candidates **4g**, **4j**, **4d** and **4k** using the bioavailability radar plots, the BOILED-EGG chart and Lipinski’s rule of five filter provided a first glance at the potential of these derivatives to be orally bioactive, though more in vivo investigations using animal models are needed to confirm the validity of these predictions.

## Data Availability

Data is contained within the article.

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
