# Peer review of "Synthesis and Evaluation of Some New 4H-Pyran Derivatives as Antioxidant, Antibacterial and Anti-HCT-116 Cells of CRC, with Molecular Docking, Antiproliferative, Apoptotic and ADME Investigations"

_pharmaceuticals, 2022, doi:10.3390/ph15070891_

Round 1

Reviewer 1 Report

The Authors have corrected the manuscript well. However, the modeling of physicochemical properties of moecules without any experimental work is highly speculative. Molecular modeling studies are expressed clearly, however there is no discussion of these results and their meaning in the larger picture. CDK2 is a well known structure and there is  plenty of literature for refering and discussing the result.

Reviewer 2 Report

The manuscript focuses on the synthesis and the biological evaluation of new 4H-pyran derivatives.

The manuscript was revised following the previous reviewers’ suggestions, in this form it can be accepted for the publication, only few issues needs to be fixed.

Details:

-          - Results and discussion: the comments on 1HNMR and 13CNMR spectra are superfluous and not relevant. Delete the sentences.

-         -  In the section of biological screening in the text it is not necessary report the sd values (eg lines 217, 221, 246, 249 etc….)

-          - Line 297: add the recent reference on the topic Future Medicinal Chemistry, 2021, 13(6), pp. 529–531

-          - Line 308: modify the list of values with the range 11-32.5

-          - A table with the IC50 values against the tested bacterial strains should be inserted in the text rather than in supplementary material.

Reviewer 3 Report

It looks like the authors implemented the all reviewer's concerns and comments on this manuscript. I have two more suggestions

1.     I suggest to the authors add figure on the first page, helping the readers to understand the background of this work.

  1. The authors didn’t include the spectrums for newly synthesized compounds in the supporting information file. Include the spectrums, so it will be helpful for the upcoming researchers.

Round 2

Reviewer 1 Report

The Authors have corrected and extended the manuscript based on previous commnets and explained the selected methods. The manuscript is at its current form a bit long and some of the figures could be placed into a supporting material if the Editors/Editorial office prefers a shorter paper.

This manuscript is a resubmission of an earlier submission. The following is a list of the peer review reports and author responses from that submission.

Round 1

Reviewer 1 Report

The first part of the manuscript deals with synthesis, antioxidant and antimicrobial studies, and is well written and well reasond. However, the study itself do not add a great value for the science because the compounds or their antibacterial activity is not novel knovledge.

In the second part, the Authors have carried out moleculae modeling studies (docking and conformational analysis). The quality of the computational study is not high enough, and the Authors should reconsider the study and its results. The study of conformers has not been described and the purpose of these calculatios is not clear. It is not clear, if the Authors have used the optimized conformes in the dockin study. From the tabulated values, it seems that the Authors have reported single point energies of the optimized  structures as some absolute, measurable energies of real molecules. However, the calculated energies do not represent any measurable energy of a molecule, but rarher an calculated value depending on used calculation theory, used force field/basis set and software. Energies should only be used in comparison of systems with equal number of atoms.

In biological studies, the Authors have used ELISA and rt-PCR to show the inhibition activity of the two most active compounds. In general, ELISA measures the amount of a protein (e.g. an enzyme) in a sample. Thus, it does not measure whether  enzyme activity is blocked or not. Thus the ELISA results shows only that the levels of CDK2 has decreased. This is most probably consequence of lowered gene-expression seen also in PCR. Expression has nothing to do with binging into enzymes binding sire. The results of molecular docking showed only that the compounds are having interactions with the binding site of a known inhibitor.

The Authors have also calculated ADME relared factors using a a web-based software. Even some of these factors can be evaluated with feasible accuracy, this kind of estimations are not reliable and should not be stated as anything else than speculations.

Altogether, I suggest the Authors to reconsider the second half of the manuscript and to find a more suitable forum for this study.

Reviewer 2 Report

The work titled "Synthesis and evaluation of some 4H-pyran derivatives as antioxidant, antibacterial and anti-HCT-116 cells of CRC. Molecular docking, quantitative measurement of CDK2 and real time PCR determination of the expression profiles of Caspase-3 and CDK genes" by Omar et al. deals with the synthesis and biological evaluation of 15 new compounds, prepared through one pot reaction. The work is not properly structured, it does not meet the standards of the journal, presenting various critical issues as described below:

The meaning of the first part of the title is unclear; it presents  English grammar errors and needs to be changed.

The abstract presents English spell errors and does not resume all the aspects addressed in the work.

The introduction does not explain the ratio used for the design of the compounds, moreover lines 61-64 are unclear and should be rewritten.

The Scheme 1 reports the synthesis of compounds 6, obtained by a two steps procedure but the in the text a one pot procedure is described and compound 5 is not cited. Moreover the synthesis involves compound 2 that is missing.

About the investigation of the cytotoxic activity, the obtained results are only preliminary and do not justify the assumption that the target of the compounds is CDK2. Further studies must be carried out.

Moreover, in Biological screening , lines 176-177 report that "the scavenging activity of all compounds was concentration-dependent" as results of table 1 should show but the reported activity is related to one concentration.

All references should be cited in the text according to the journal style.